# How distinct sources of nuisance variability in natural images and scenes limit human stereopsis

David N. White [1,2]*, Johannes Burge [1,3,4]

1 Neuroscience Graduate Group, University of Pennsylvania, Philadelphia, Pennsylvania, United States of America, 2 Department of Electrical Engineering & Computer Science, York University, Toronto, Ontario, Canada, 3 Department of Psychology, University of Pennsylvania, Philadelphia, Pennsylvania, United States of America, 4 Department of Bioengineering, University of Pennsylvania, Philadelphia, Pennsylvania, United States of America

* dave.cni.upenn@gmail.com

## Abstract

Stimulus variability—a form of nuisance variability—is a primary source of perceptual uncertainty in everyday natural tasks. How do different properties of natural images and scenes contribute to this uncertainty? Using binocular disparity as a model system, we report a systematic investigation of how various forms of natural stimulus variability impact performance in a stereo-depth discrimination task. With stimuli sampled from a stereo-image database of real-world scenes having pixel-by-pixel ground-truth distance data, three human observers completed two closely related double-pass psychophysical experiments. In the two experiments, each human observer responded twice to ten thousand unique trials, in which twenty thousand unique stimuli were presented. New analytical methods reveal, from this data, the specific and nearly dissociable effects of two distinct sources of natural stimulus variability—variation in luminance-contrast patterns and variation in local-depth structure—on discrimination performance, as well as the relative importance of stimulus-driven-variability and internal-noise in determining performance limits. Between-observer analyses show that both stimulus-driven sources of uncertainty are responsible for a large proportion of total variance, have strikingly similar effects on different people, and—surprisingly—make stimulus-by-stimulus responses more predictable (not less). The consistency across observers raises the intriguing prospect that image-computable models can make reasonably accurate performance predictions in natural viewing. Overall, the findings provide a rich picture of stimulus factors that contribute to human perceptual performance in natural scenes. The approach should have broad application to other animal models and other sensory-perceptual tasks with natural or naturalistic stimuli.

## Author summary

Linking properties of the external world, and of sensory stimuli, to how neurons and animals respond has proven an important approach to understanding how the brain works. Much is

**Data availability statement:** All data used in this study are available in CSV format on a

GitHub repository at https://github.com/burgelab/WhiteBurge2025-Data MATLAB code used to perform analyses is available on a repository at https://github.com/burgelab/WhiteBurge2025-Code.

**Funding:** This research was funded by a Research Project Grant from the US NIH National Eye Institute and Office of Behavioral and Social Sciences (Grant number R01-EY028571 to JB; https://www.nei.nih.gov/ and https://obssr.od.nih.gov/) and a Training Grant from the US NIH National Eye Institute (Grant number 5T32EY007035-40 to DNW; https://www.nei.nih.gov/). The funders had no role in study design, data collection and analysis, decision to publish, or preparation of the manuscript.

**Competing interests:** The authors have declared that no competing interests exist.

known about how nervous systems respond to simple stimuli. Less is known about how systems respond to real-world stimuli. A major challenge is to present stimuli in an experimental setting that reflect important aspects of the variability that is present in natural viewing, while maintaining the rigor and interpretability that is necessary for drawing scientific conclusions about what drives and limits perceptual performance. Using a high-fidelity database of natural images and scenes, we conducted two human stereo-depth discrimination experiments and analyzed the data with a newly developed method that reveals how distinct features of natural scenes and images impact performance. Results show that stimulus-by-stimulus variation has highly consistent effects on different people. The approach should have broad application to other animal models and other sensory-perceptual tasks.

## Introduction

An ultimate goal for perception science is to understand and predict how perceptual systems work in the real world. One approach to achieving this goal is to probe the system with naturalistic stimuli—stimuli that are derived from the natural environment, or bear substantial similarities to them. By examining how stimulus variation characteristic of real-world scenes affects stereo-depth discrimination, we show that performance patterns are similar across different humans, and we partition the effects of distinct stimulus and scene factors on performance—with some surprising results. Further, natural-stimulus variation causes a high degree of stimulus-by-stimulus consistency across observers, consistency that, in principle, could be used to develop and constrain future image-computable models of human perceptual performance.

There is a long tradition of investigating visual performance in human and animal models using simple stimuli and simple tasks. Recent years have been marked by the realization that simple stimuli and tasks may be insufficiently complex to understand how vision works in the real world. A number of recent efforts have taken steps to make the tasks during which psychophysical and neurophysiological data are collected more ecologically valid, while using traditional stimuli (e.g. gratings, Gabors). Some such efforts have, for example, removed the requirement that animals maintain fixation, allowing them fixate freely on stimuli presented on a monitor [1]. Here, we use a traditional forced-choice task, and focus effort on probing perceptual performance with stimuli that are more similar to those encountered in real-world viewing situations (see Discussion).

The use of natural or naturalistic stimuli, however, poses challenges. With such stimuli, it is difficult to maintain the rigor and interpretability that has characterized classic research. One important source of difficulty is the sheer number of factors that inject variability into natural retinal images. Some of these factors depend on the environment: the textural patterns on surfaces, the 3D structure of those surfaces, and how the objects that own those surfaces are arranged in 3D space. Other factors are due to the organism and its relationship to the environment, including the optical state of the eyes and the posture and movements of the eyes, head, and body relative to objects in the scene. All of these factors combine to generate many different retinal images, all of which are associated with a particular value of a distal property (e.g. depth) of interest. Such natural-stimulus variability—a form of "nuisance stimulus variability"—impacts neural response [2–4], and is an important reason that estimation and discrimination of behaviorally-relevant latent variables (e.g. depth, size, 3D orientation) is difficult. In order to perform well, perceptual systems must select for proximal stimulus features that provide information about the latent variable of interest, while generalizing across (i.e. maintaining invariance to) stimulus variation that is not useful to the task. In natural viewing, the computations run by the vision system should minimize, to the maximum

possible extent, the degree to which natural-stimulus variability causes variability in human estimation and discrimination in each critical task [5–13].

Using binocular disparity as a model system, we report a systematic investigation of how various forms of natural-stimulus variability impact performance in a depth discrimination task. To approximate natural-stimulus variation, thousands of stimuli were sourced from a natural stereo-image database with co-registered laser-based range data at each pixel using constrained sampling techniques. The sampled stimuli were used to probe human depth-from-disparity discrimination and to determine distinct properties of natural scenes that place limits on human performance. With appropriate experimental designs and data-analysis methods, the natural (random) variation across the uncontrolled aspects of the stimuli in each condition provides one with the ability to determine the limits that distinct types of nuisance stimulus variability place on performance.

Two experiments were conducted using the double-pass psychophysical paradigm [13–16]. In contrast to typical 2AFC forced-designs, in which hundreds of responses are collected for each unique stimulus (or trial), double-pass experiments collect two responses for each of two presentations of hundreds of unique stimuli (or trials) in each condition. The conditions of the experiments were defined by different fixation disparities and levels of local-depth variation. These aspects of the stimuli were parametrically manipulated and tightly controlled. Other aspects of the stimuli—luminance-contrast patterns and local-depth structure—were allowed to vary randomly (as they do in natural viewing). We develop new analytical methods that allow us to infer, from the double-pass data, (i) the relative importance of natural-stimulus variability and internal noise in limiting performance, and (ii) the specific impact that distinct sources of natural-stimulus variability—luminance-pattern variability and local-depth variability—have on performance.

Several key findings emerge. First, we replicate a performance pattern from the classic literature: discrimination thresholds increase exponentially as targets move farther in depth from fixation. Second, we show that performance limits are increasingly attributable to stimulus variability (rather than internal noise) as the stimuli used to probe performance have more local-depth variability. Third, we show that two distinct types of natural-stimulus variability—luminance-pattern variation and local-depth variation—have distinct and largely separable effects on human performance. Fourth, we find that as stimulus variation becomes more severe, the absolute impact of that stimulus-by-stimulus variation on performance becomes more severe and also becomes more uniform across human observers.

## Materials and methods

### Ethics statement

All observers provided informed written consent in accordance with the declaration of Helsinki. The Institutional Review Board at the University of Pennsylvania approved all protocols and experiments.

### Human observers

All observers had normal or corrected-to-normal acuity. Two of the observers were authors, and the third was naive to the purpose of the study.

### Data and software

Psychophysical experiments were performed in MATLAB 2017a using Psychtoolbox version 3.0.12. Stimulus sampling and data-analyses were also performed in MATLAB 2017a. Data

are available at https://github.com/burgelab/WhiteBurge2025-Data. Code used to perform the analyses is available at https://github.com/burgelab/WhiteBurge2025-Code.

## Apparatus

Stimuli were presented on a custom-built four-mirror haploscope. The haploscope displays were two identical VPixx ViewPixx 23.9 inch LED monitors. Displays were 53.3 × 30.0 cm in size, with 1920 × 1080 pixel resolution and a native 120 Hz refresh rate. The maximum luminance of each display was 106 cd/m$^2$. After light loss due to mirror reflections, the effective luminance was 94 cd/m$^2$. The mean background gray level of the displays was set to 40 cd/m$^2$. The gamma function was linearized over 8 bits of gray level.

All mirrors in the haploscope were front-surface mirrors, to eliminate secondary reflections. The mirrors most proximal to the observer were housed in mirror cubes with 2.5 cm circular viewports. The viewports were positioned 65 mm apart, a typical human interpupillary distance. The openings of the cubes limited the field of view to approximately 16° of visual angle.

The optical and vergence distances of the displays were set to 1.0 m. This distance was verified both by standard binocular sighting techniques and via laser distance measurement. At this distance, each pixel subtended 1.07 arcmin. A chin and forehead rest stabilized the head of each observer.

## Stimuli

Stereo-image patches (32 × 32 pixels each for the left- and right-eye patches) were sampled from 98 large stereo-images (1920 × 1080 pixels) of the natural environment with co-registered laser range data at each pixel [17]. Sampling procedures are described below. Image patches were presented dichoptically and subtended 1° of visual angle. The center pixel of the stereoscopically-specified scene patch was located straight-ahead along the observer's cyclopean line of sight and had uncrossed disparity with respect to the display. The patch was spatially windowed by a raised-cosine function—a Hann window function—having zero disparity with respect to the display. When viewed binocularly, the patch of scene appeared in depth behind a fuzzy aperture; the aperture appeared in the depth-plane of the display. When viewed monocularly, the patch appeared to fade into the mean luminance surround. Uncrossed fixation disparities (i.e. uncrossed disparity pedestals) were introduced at the stereo-patch sampling stage by cropping the patch from its source image, assuming that a virtual pair of eyes was fixating a point along the cyclopean line of sight in front of the sampled scene location (i.e. a virtual fixation error) [18]. The size of the virtual fixation error was set such that the uncrossed disparity would have the desired value when the stereo-patch was viewed in the haploscope rig.

Each stereo-image patch in the dataset was labeled by the amount of local-depth variation in the imaged scene region, as quantified by disparity-contrast. Disparity-contrast is given by the root-mean-squared difference between the vergence demand of the central corresponding point and the vergence demands of the points in the local surround

$$c_\delta = \sqrt{\frac{\sum\limits_{\mathbf{x}} (\mathbf{v}(\mathbf{x}) - \mathbf{v}_0)^2 \, \mathbf{w}(\mathbf{x})}{\sum\limits_{\mathbf{x}} \mathbf{w}(\mathbf{x})}}, \tag{1}$$

where $\mathbf{v}_0$ is the vergence angle that is required to fixate the 3D-scene point specified by the center pixels of the left- and right-eye image patches, $\mathbf{v}(\mathbf{x})$ is the vergence angle required to fixate the scene points corresponding to the other pixels in the patch, $\mathbf{w}$ is the raised cosine window function, and $\mathbf{x} = \{x, y\}$ is the spatial location of each pixel. Note that the difference in vergence demand $\mathbf{v}(\mathbf{x}) - \mathbf{v}_0$ is simply equal to the relative disparity between the center pixel and the other pixels in the patch. The vergence demand at each point in the patch was computed for an observer viewing the stimulus at the viewing distance and direction set by the experimental rig (i.e. 1 meter away, straight-ahead). Note that that $\mathbf{v}_0$ is identical for all stimuli having the same nominal disparity, but that the differences in vergence demand $\mathbf{v}(\mathbf{x}) - \mathbf{v}_0$ (i.e. the pattern of relative disparity) is unique to each stimulus.

Each stereo-image patch was contrast fixed to the median root-mean-squared (RMS) contrast (i.e. $c_{\mathrm{rms}} = 0.3$) in the natural-stimulus dataset. RMS contrast is given by

$$c_{\mathrm{rms}} = \sqrt{\frac{\sum\limits_{\mathbf{x}} \left( \mathbf{c_L}^2(\mathbf{x}) + \mathbf{c_R}^2(\mathbf{x}) \right) \mathbf{w}(\mathbf{x})}{\sum\limits_{\mathbf{x}} \mathbf{w}(\mathbf{x})}}, \tag{2}$$

where $\mathbf{c}_L$ and $\mathbf{c}_R$ are the left- and right-eye Weber contrast image patches, $\mathbf{w}$ is the window function, and $\mathbf{x} = \{x, y\}$ is the location of a given image pixel.

**Stimulus sampling** Left- and right-eye image patches were sampled from a natural image database with pixel-wise co-registered range data [17]. Because the stereo-photographs were of natural scenes, each local patch was characterized by a different luminance pattern and by some amount of local-depth variability. Corresponding points in the image were determined directly from the range data (see [18]). Sampled patches were then cropped such that each resulting stimulus patch had specified fixation disparities (i.e. pedestal disparities) relative to the corresponding point (see Stimuli above). Patches were screened to ensure that the disparity variability within the central $1/8°$ ($\approx 4$ pixel diameter) region of each patch equaled the nominal fixation disparity within a tight tolerance (see below). Note that because depth varies naturally across any given patch, this central region was the only region of the patch that was guaranteed to equal the nominal fixation disparity. Because we were interested in the effect of disparity-contrast on performance, we sampled patches whose disparity-contrast, when viewed in the experimental rig, fell into a "low" range (0.025–0.117 arcmin) or a "high" range (0.393–1.375 arcmin). We report performance below for patches having each disparity-contrast separately. To ensure that each stereo-image patch was unique, patches were not allowed to overlap radially in their source images by more than 10 pixels; this level of overlap was rare.

If the viewing geometry (i.e. distance and direction) of stimulus presentation in an experimental rig does not match the viewing geometry during stereo-image patch sampling, the stereo-specified 3D structure of presented stimulus will be distorted relative to the geometry of the original scene [19]. Stereo-image patches were sampled from all distances and directions, but presented patches at a fixed distance and direction (i.e. one meter away, straight-ahead). Hence, the stereo-specified depth structure during presentation was distorted from that in the original 3D scene. It is possible to prevent these distortions, but only at the cost of distorting the left- and right-eye luminance images. We opted to preserve luminance structure rather than the details of the stereo-specified 3D geometry of the original natural scene. Throughout the article, the disparity-contrast values that are used to characterize the stereo-specified depth variation in each stereo-image patch were set by each patch as it was viewed by the participants in the experimental rig.

**Stimulus vetting** Before being included in the experimental stimulus set, stereo-image patches underwent a vetting procedure. The vetting procedure had two primary aims.

The first, most fundamental aim was to ensure accurate co-registration between the luminance and range information in the half-images of each patch. Accurate co-registration was critical for all aspects of the experiment, because the values of the independent variables (i.e. disparity and disparity-contrast) are determined directly from the range data. Although inaccurate co-registration was rare, it was present in a non-negligible proportion of patches. In such cases, the luminance data that observers would have used to estimate disparity would have been inconsistent with the range data used by the experimenters to compute the nominal ground truth disparity. Hence, failing to identify and exclude poorly co-registered patches would mar the accuracy of the results. Potential stereo-image patches were manually vetted by viewing each patch in the experimental rig with onscreen disparities that were nominally uncrossed, zero, or crossed with respect to the screen. Patches that did not pass scrutiny (i.e. that had the wrong depth relationship relative to the screen) were discarded from the pool. The manual vetting procedure was conducted until thousands of unique stimulus patches without co-registration problems were obtained.

The second aim of the vetting procedure—which was enforced programmatically—was to ensure that the center of each stereo-image patch was a coherent target for depth estimation (see above). We required that the most central region of each patch contained neither a substantial change in disparity (i.e. a disparity-contrast greater than 20 arcsec), or a half-occluded region. Pixels containing half-occluded regions were allowed outside of the most central region. Because regions that are half-occluded have undefined disparity, stimuli including a half-occluded region have undefined disparity-contrast. For patches containing half-occlusions, disparity-contrast was computed by excluding pixels corresponding to half-occluded regions of the scene from the calculation. We did not exclude stimuli with half-occlusions from the dataset because they occur commonly in natural viewing [18].

**Stimulus flattening** From the sampled set of natural stereo-image patches—which contain both natural luminance-pattern variation and natural-depth variation—we also created a "flattened"—but otherwise matched—dataset of stereo-image patches. To convert patches with natural-depth structure into patches with flat depth structure, either the left- or right-eye half-image patch (chosen randomly) was replaced by a duplicate of the remaining right- or left-eye half-image patch. This procedure ensured that there is essentially zero-disparity variation across the patch, such that the disparity pattern specifies a fronto-parallel plane.

## Procedure

Stimuli were presented at the center of a fixation crosshairs. The crosshairs were positioned in the center of a circular, 4° diameter, mean-luminance gray area. The circular area was surrounded by a mean-luminance 1/f noise field. The crosshairs consisted of a 2° diameter circle punctuated by hairs jutting outwards at the cardinal and diagonal directions. Hairs were 1° in length and 4.2 arcmin in thickness.

Stimuli were presented using a two-interval forced choice (2IFC) procedure. Each interval had a duration of 250 ms. The inter-stimulus interval was also 250 ms. In one interval of each trial, a stimulus with a standard disparity was presented. In the other interval, a stimulus with a comparison disparity was presented. The order in which the standard or comparison stimulus was presented was randomized. All stimuli (standards and comparisons) were always unique across all intervals and trials of an experiment, having been sampled from different locations and scenes across the source image set.

The task was to report, with a key press, whether the stimulus in the second interval appeared to be nearer or farther than the stimulus in the first interval. Feedback was provided after each response: a high frequency tone indicated a correct response; a low frequency tone indicated an incorrect response.

Psychometric data were collected in a fully-crossed design with disparity pedestal and disparity-contrast as the independent variables. For each combination of disparity pedestal and disparity-contrast, the method of constant stimuli was used for stimulus presentation. Disparity pedestals were defined by one of five standard disparities: $\delta_{std}$ = $[-11.25, -9.38, -7.5, -5.63, -3.75]$ arcmin. Five equally spaced comparison disparities were paired with each standard. Disparity-contrast levels were defined as $\delta_C$ = $[0.025—0.117, 0.393—1.375]$ arcmin, which were labeled "low" and "high" disparity-contrasts respectively. Stimuli in the low disparity-contrast conditions were just-noticeably non-flat to observers. Stimuli in the high disparity-contrast conditions appeared quite noticeably non-uniform in depth. The high disparity-contrast condition contained stimuli that were easily fusible in most cases.

The disparity-contrast levels and the comparison disparities in each condition were chosen based on pilot data. Comparison disparities were chosen so that data points on the psychometric function ranged from 10% to 90% in the low disparity-contrast condition. Data points at 0% and 100% provide no useful information for estimating decision variable correlation (see "Partitioning the variability of the decision variable" section). Before collecting the data, each observer completed practice sessions to ensure that discrimination performance was stable.

To simulate the stimulus variability that occurs in natural-viewing conditions, a unique natural stereo-image patch was presented on each interval of each trial. This feature of the experimental design represents a departure from more standard experimental designs, in which either the same stimulus is presented many times each or stimulus differences (e.g. different random dot stereograms) are considered unimportant and not analyzed.

Experiments were conducted using a double-pass experimental paradigm. In double-pass experiments, observers respond to the exact same set of unique trials two times each. Double-pass experiments enable one to determine the relative importance of factors that are repeatable across trials (e.g. external stimulus variation), and factors that vary randomly across trials (e.g. internal noise).

Two double-pass experiments were conducted. In one, all stimuli had natural-depth variation. In the other, all stimuli were "flattened" (see "Stimulus flattening" section). Importantly, both double-pass experiments used the same scene-locations (and hence, near-identical luminance contrast patterns). This design feature allowed us to examine the relative importance of luminance-pattern-driven variability and disparity-contrast-driven variability in the decision variable (see "Partitioning the externally-driven component of the decision variable" section).

Over the course of each double-pass experiment, 10,000 unique stimuli were presented in 5000 unique trials of each double-pass experiment. Five-hundred trials were collected in each of ten conditions (5 standard disparities ×2 disparity-contrasts). Data were collected in 100-trial blocks (i.e. twenty repeats per comparison disparity level per block). The order in which the blocks were run was randomized and counterbalanced across conditions. Two double-pass experiments were conducted, for a total of 20,000 trials per observer.

## Psychometric fitting

Cumulative Gaussian functions were fit to the psychometric data in each condition using maximum-likelihood methods. Discrimination thresholds were calculated from the fitted

functions. The relationship between the sensitivity index $d'$ (i.e. d-prime) and percent the comparison chosen PC in a two-interval two-alternative forced-choice experiment is given by

$$PC = \Phi\left(\frac{d'}{\sqrt{2}}\right), \tag{3}$$

where $\Phi$ is the cumulative normal function, with $d'$ given by

$$d' = \frac{\Delta\delta}{\sigma_T}, \tag{4}$$

where $\Delta\delta = \delta_{cmp} - \delta_{std}$ is the difference between the comparison and standard disparities (i.e. the mean value of the decision variable), and $\sigma_T^2$ is the variance of the underlying decision variable. (In accordance with standard practices, we assume that decision variable variance is constant for all comparison-disparity levels at a given standard-disparity level—that is, pedestal disparity. The psychometric data are consistent with this assumption.)

The discrimination threshold $T$ is set by choosing a criterion d-prime that defines the just-noticeable difference. In a two-interval, two-alternative forced-choice (2AFC) experiment, threshold is given by

$$T = \sqrt{\sigma_T^2\, d'_{crit}}, \tag{5}$$

where $d'_{crit}$ is the criterion d-prime. For computational simplicity, we assume a criterion d-prime of 1.0 such that threshold level performance corresponds to the 76% point on the psychometric function. Thresholds are thus given by the change in the disparity required to go from the 50% to the 76% points on the psychometric function.

Discrimination thresholds were computed from data across both passes of the experiment. When fitting psychometric data across one or both double-pass experiments (see below), thresholds were constrained to change log-linearly across disparity pedestals. Under this constraint, discrimination thresholds in the conditions of a double-pass experiment associated with a given disparity-contrast are specified by

$$T = \sigma_T = \exp(m\delta_{std} + b), \tag{6}$$

where $\delta_{std}$ is the standard pedestal disparity, $m$ and $b$ are the slope and y-intercept of the line characterizing the log-thresholds. This constraint is consistent with the predictions of normative models of disparity discrimination with natural stimuli previously reported patterns in psychophysical data [20] and with the log-linear patterns in the current threshold data (see "Experiment 1" and "Experiment 2" subsections of the Results section below). The maximum-likelihood estimates of the parameters defining threshold under the constraint were fit across all conditions having a given disparity-contrast. They are given by

$$\hat{m}, \hat{b} = \arg\max_{m,\, b} \sum_s L_s\big(\big[\exp(m\delta_{std}^{(s)} + b)\big]^2\big), \tag{7}$$

where $L_s$ is the likelihood of the raw response data in the $s$th condition, under the assumption that percent correct is governed by a cumulative normal function with mean parameter equal to the $s$th disparity pedestal $\delta_{std}^{(s)}$ and variance parameter equal to $\big[\exp(m\delta_{std}^{(s)} + b)\big]^2$.

Finally, the variance of decision variable at each pedestal disparity was obtained by plugging these estimated parameters into Eq (6).

## Modeling the decision variable

The decision variable can be modeled as a difference between disparity estimates from the stimuli presented on each trial

$$D = \hat{\delta}_{\text{cmp}} - \hat{\delta}_{\text{std}}, \tag{8}$$

where $\hat{\delta}_{\text{std}}$ is the estimate from the stimulus with the standard disparity and $\hat{\delta}_{\text{cmp}}$ is the estimate from the stimulus with the comparison stimulus. In accordance with signal detection theory, if the value of the decision variable is greater than zero (and if the observer sets the criterion at zero), the observer will select the stimulus with the comparison disparity. If the decision variable is less than zero, the observer will select the stimulus with the standard disparity.

The decision variable can be more granularly modeled as the sum of two independent random variables. The first random variable accounts for stimulus-driven variability (i.e. variance that is due to nuisance stimulus variability), and has its value set by the particular stimulus (or stimuli) that are presented on a given trial. The second random variable accounts for internal noise, and has its value set randomly on each trial. In a double-pass experiment, across the two presentations of a particular unique trial in a double-pass experiment (i.e. the presentation in the first pass and the presentation in the second pass), the value of the decision variables are given by

$$\begin{aligned} D_1 &= V + W_1, \\ D_2 &= V + W_2, \end{aligned} \tag{9}$$

where $V$ is stimulus-driven contribution to the decision variable, $W$ is a sample of internal noise, and the subscripts index on which pass the trial was presented. Across the two passes of the double-pass experiment, the decision variables can be described as a single two-dimensional random variable $\mathbf{D} = [D_1, D_2]^\mathsf{T}$.

The stimulus-driven component of the decision variable on a single pass of the experiment $V \sim \mathcal{N}(\delta_{\text{cmp}} - \delta_{\text{std}},\ \sigma_E^2)$ is modeled as unbiased and normally distributed with stimulus-driven variance $\sigma_E^2$. The noise-driven component of the decision variable $W \sim \mathcal{N}(0, \sigma_I^2)$ is modeled as zero-mean and normally distributed with variance $\sigma_I^2$. If the external (i.e. stimulus-driven) and internal (i.e. noise-driven) components of the decision variable are independent, as we assume they are here, the total variance of the decision variable on a given pass is given by the sum of the internal and external components

$$\sigma_T^2 = \sigma_E^2 + \sigma_I^2. \tag{10}$$

## Decision-variable correlation

The correlation of the decision variable across passes is given by the fraction of the total variance that is accounted for by external (i.e. stimulus-driven) factors, the factors that are repeated across passes. Hence, decision-variable correlation is given by

$$\rho = \frac{\sigma_E^2}{\sigma_T^2} = \frac{\sigma_E^2}{\sigma_E^2 + \sigma_I^2}, \tag{11}$$

where $\sigma_E^2$ is the component of the decision-variable variance accounted for by external (i.e. stimulus-driven) factors and $\sigma_I^2$ is the component of the decision-variable variance accounted for by internal factors (i.e. noise). (Note that decision variable correlation Eq (11) should not be confused with variance accounted for ($R^2$), a statistic that is often computed in regression analyses.) In order to partition stimulus- and internally-driven sources of variability, we combine estimates of decision-variable correlation and discrimination thresholds (see below). Decision-variable correlation is an integral factor in determining the repeatability of observer responses across passes of a double-pass experiment.

**Estimating decision-variable correlation**  Decision-variable correlation was estimated via maximum likelihood from the pattern of observer response agreement between passes. The log-likelihood of $n$-pass response data, under the model of the decision variable, is

$$\mathcal{L}_n(\boldsymbol{\theta}) = \sum_j N_j \, \log \mathcal{L}_n^j(\boldsymbol{\theta}), \tag{12}$$

where $\boldsymbol{\theta}$ represents the parameter(s) to be estimated, $j$ is a specific pattern of response, $N_j$ represents the number of times a specific pattern of response was measured. For a double-pass experiment ($n = 2$), the set of response patterns are given by the combination of all possible combinations of responses for each pass. The number of patterns of binary responses is $N = 2^n$. For 2IFC experiment, $N = 2^2 = 4$, with patterns of responses $j \in \{[-,-], [-,+], [+,-], [+,+]\}$. Here, we use $+$ to indicate that the comparison was chosen and $-$ indicates the comparison was not chosen.

We model the joint decision variable as a vector drawn from a multivariate normal distribution $\mathbf{D} \sim \mathcal{N}(\mathbf{x}; \mathbf{m}, \Sigma)$ with a mean vector $\mathbf{m}$ and covariance matrix $\Sigma$. The likelihood of a particular pattern of response is given by

$$\mathcal{L}_2^j(\boldsymbol{\theta}) = \int_{s^j(c_1, c_2)} \mathcal{N}(\mathbf{x}; \mathbf{m}, \Sigma) \, d\mu(\mathbf{x}), \tag{13}$$

where integration is in respect to probability measure $\mu$ and $s^j$ is a subset of the support $S$. Here, $s^j$ defines the integration limits for a specific pattern of response $j$ and is a function of the decision criterion on each pass $c \in \{c_1, c_2\}$. Specifically, the integration limits for each dimension/pass are determined by the values of response pattern. For a response $r_i$ at pass $i$, where the comparison is not chosen ($r_i = -$), $P(D_i < c_i)$ and the integration limits are $[-\infty, c_i]$. Likewise, for comparison chosen ($r_i = +$), $P(D_i \geq c_i)$ with integration limits $[c_i, \infty]$.

It is computationally convenient to estimate decision-variable correlation with a normalized joint decision variable $\mathbf{D}^z = [D_1^z, D_2^z]^\mathsf{T}$ such that it has unit variance on each pass. Normalizing the joint decision variable sets the normalized means equal to $d'$. Normalizing the joint decision variable also confers a practical advantage in converting the covariance matrix into a correlation matrix so that it can be fully characterized by decision-variable correlation.

The normalized mean vector and normalized covariance (i.e. correlation) matrix associated with the normalized joint decision variable are given by $\mathbf{m}^z = \mathrm{M}\mathbf{m}$, and $\Sigma^z = \mathrm{M}\Sigma\mathrm{M}$, where the superscript $z$ indicates a normalized parameter, and $\Sigma^z$ is the correlation matrix (i.e. the covariance matrix of the normalized joint decision variable). The normalizing matrix is given by $\mathrm{M} = \mathrm{diag}(\frac{1}{\boldsymbol{\sigma}_T})$, where $\boldsymbol{\sigma}_T$ is a vector of the standard deviation of the joint decision variable $\mathbf{D}$ in each pass, and where the $\mathrm{diag}(\cdot)$ function converts a vector into a matrix with the

vector-values on the diagonal. The correlation matrix is given by

$$\Sigma^z = \begin{bmatrix} 1 & \rho \\ \rho & 1 \end{bmatrix}. \tag{14}$$

Substituting parameters associated with the normalized decision variable into equations, yields mathematically equivalent expressions of the likelihoods:

$$\mathcal{L}_2^j(\boldsymbol{\theta}) = \int\limits_{s^j(c_1^z, c_2^z)} \mathcal{N}\left(\mathbf{x}^z; \mathbf{m}^z, \Sigma^z\right) \, d\mu(\mathbf{x}^z). \tag{15}$$

We also assume that the criteria associated with the normalized decision variable on all passes equals zero, which is justified by the data and by the two-interval, two-alternative forced choice design. In the general case, when this assumption is not made, the decision criteria should also be normalized—that is, the normalized criteria are given by $\mathbf{c}^z = \mathbf{Mc}$. Thus, when analyzing double-pass experimental data under the indicated assumptions, decision variable correlation $\boldsymbol{\theta} = \rho$ is the only parameter that needs to be estimated. Specifically, the maximum-likelihood estimate of decision variable correlation is given by

$$\hat{\rho} = \arg\max_{\rho} \sum_j N_j \, \log \mathcal{L}_2^j(\rho), \tag{16}$$

where $N_p \in \{N_{--}, N_{-+}, N_{+-}, N_{++}\}$ is the number of each type of response agreement or disagreement, and $\mathcal{L}_p \in \{\mathcal{L}_{--}, \mathcal{L}_{-+}, \mathcal{L}_{+-}, \mathcal{L}_{++}\}$ is the likelihood of the data given an underlying decision variable distribution specified by the decision variable correlation. The likelihoods are given by

$$
\begin{aligned}
\mathcal{L}_2^{--} &= \int_{-\infty}^{c_1^z} \int_{-\infty}^{c_2^z} \mathcal{N}\left(\mathbf{x}^z; \mathbf{m}^z, \Sigma^z\right) \, d\mu(x_1, x_2), \\
\mathcal{L}_2^{-+} &= \int_{-\infty}^{c_1^z} \int_{c_2^z}^{\infty} \mathcal{N}\left(\mathbf{x}^z; \mathbf{m}^z, \Sigma^z\right) \, d\mu(x_1, x_2), \\
\mathcal{L}_2^{+-} &= \int_{c_1^z}^{\infty} \int_{-\infty}^{c_2^z} \mathcal{N}\left(\mathbf{x}^z; \mathbf{m}^z, \Sigma^z\right) \, d\mu(x_1, x_2), \\
\mathcal{L}_2^{++} &= \int_{c_1^z}^{\infty} \int_{c_2^z}^{\infty} \mathcal{N}\left(\mathbf{x}^z; \mathbf{m}^z, \Sigma^z\right) \, d\mu(x_2, x_2).
\end{aligned}
\tag{17}
$$

The decision-variable correlation analyses presented in this work were computed under the assumption that the means were equivalent and criteria were zero. Under these assumptions, the two likelihoods of response disagreements are equal ($\mathcal{L}_2^{-+} = \mathcal{L}_2^{+-}$), thus simplifying the computation of Eq (17). However, we first verified that these assumptions were reasonable for the current dataset.

**Determining the variances of the decision-variable components** With an estimate of the total variance of the decision variable and an estimate of decision-variable correlation, one can estimate the variances of the externally- and internally-driven components of the decision

variable. Plugging Eq (5) into Eq (11) and rearranging yields an estimate of the variance the externally-driven component of the decision variable

$$\hat{\sigma}_E^2 = \hat{\rho}\hat{\sigma}_T^2. \tag{18}$$

Plugging this estimate into Eq (10) and rearranging gives an expression for the internally-driven component of the decision variable

$$\hat{\sigma}_I^2 = \hat{\sigma}_T^2 - \hat{\sigma}_E^2. \tag{19}$$

This series of analytical steps was performed for the two double-pass experiments that were conducted: one with natural and one with flattened local-depth variation.

## Partitioning the externally-driven component of the decision variable

To estimate the contributions of luminance-pattern- and local-depth-driven (i.e. disparity-contrast-driven) variability to the decision variable, performance was compared across the stimulus sets with natural and flattened local-depth variation. Recall that the flattened stimulus set effectively eliminates local-depth variability from the natural-stimulus set—because the disparity pattern in each flattened stimulus specifies a fronto-parallel plane—while leaving luminance-contrast patterns essentially unaffected. Hence, because the luminance-pattern-driven component should be essentially the same in both stimulus sets, and because the local-depth-driven component is eliminated in one of the two stimulus sets, an appropriate comparison should reveal the impact of each factor.

To compare performance across the flattened and natural-stimulus sets, we simultaneously analyzed all data from both double-pass experiments using a quasi-quadruple-pass analysis (see below).

**Expanded decision variables and correlations**  Before explaining in detail how to estimate the contribution of two distinct stimulus-driven factors it is necessary to show how the decision variable depends on these factors in each of the two double-pass experiments. The decision variables in the experiments with flattened and natural-stimuli are given, respectively, by

$$V_\dagger = L, \tag{20}$$

$$V_* = L + B, \tag{21}$$

where $L$ and $B$ denote the the luminance-pattern- and local-depth-driven components of the decision variable, respectively, and $\dagger$ and $\star$ indicate, respectively, whether the decision variable corresponds to stimuli that have been flattened (2nd double-pass experiment) or have natural-depth profiles (1st double-pass experiment). (Note that, for the simplicity of mathematical development, we present the equations here in the Methods section in the opposite order from which the experiments were conducted and presented in the Results section).

Plugging these expanded forms for the externally-driven component of the decision variable into Eq (9) yields expanded expressions for the decision variables in each of the two

double-pass experiments

$$
\begin{aligned}
D_\dagger &= \overbrace{(L\quad\quad)}^{V_\dagger} + W_\dagger, \\
D_* &= \underbrace{(L + B)}_{V_*} + W_*.
\end{aligned}
\tag{22}
$$

Clearly, the presence or absence of the local-depth-driven component of the decision variable was the only component that differed across the two double-pass experiments.

Decision-variable correlations across passes in the flattened and natural double-pass experiments, in terms of these new variables, are given by

$$
\begin{aligned}
\rho_{\dagger\dagger} &= \frac{\sigma^2_{E\dagger}}{\sigma^2_{T\dagger}} = \frac{\sigma^2_L}{\sigma^2_L + \sigma^2_{I\dagger}}, \\
\rho_{**} &= \frac{\sigma^2_{E*}}{\sigma^2_{T*}} = \frac{\sigma^2_L + \sigma^2_B + 2\mathrm{cov}[L,B]}{\sigma^2_L + \sigma^2_B + 2\mathrm{cov}[L,B] + \sigma^2_{I*}},
\end{aligned}
\tag{23}
$$

where $\sigma^2_{T\dagger}$ and $\sigma^2_{T*}$ are variabilities of the decision variable, where $\sigma^2_L$ and $\sigma^2_B$ are the luminance-pattern and local-depth-driven contributions to response variability, $\sigma^2_{I\dagger}$ is the internal noise when only luminance-pattern-driven variability is present, $\sigma^2_{I*}$ is the internal noise when both luminance-pattern- and local-depth-driven variability is present, $\dagger$ indicates comparisons across between passes in the double-pass experiment with flattened-depth profiles (i.e. the second double-pass experiment), and $**$ indicates comparisons across passes in the double-pass experiment with natural-depth profiles (i.e. the first double-pass experiment). Clearly, there are five unknowns—$\sigma^2_L$, $\sigma^2_B$, $\mathrm{cov}[L,B]$, $\sigma^2_{I\dagger}$, and $\sigma^2_{I*}$—and, including the threshold equations from each of the two double-pass experiments (see Eqs (5) and (10)), but only four equations. However, by computing decision-variable correlation between passes across each of the two double-pass experiments, a fifth equation is obtained. Specifically,

$$
\rho_{\dagger*} = \frac{\sigma^2_L + \mathrm{cov}[L,B]}{\sigma_{T\dagger}\sigma_{T*}} = \frac{\sigma^2_L + \mathrm{cov}[L,B]}{\sqrt{\left(\sigma^2_L + \sigma^2_{I\dagger}\right)}\sqrt{\left(\sigma^2_L + \sigma^2_B + 2\mathrm{cov}[L,B] + \sigma^2_{I*}\right)}},
\tag{24}
$$

where $\dagger*$ indicates the cross-double-pass-experiment comparisons. Now, with five equations and five unknowns, the equations can be solved.

**Estimating decision-variable correlation with expanded decision variables**  A novel quasi-quadruple-pass analysis was used to simultaneously estimate $\rho_{\dagger\dagger}$, $\rho_{**}$, and $\rho_{\dagger*}$, the decision-variable correlations across all four passes of the two double-pass experiments. The quasi-quadruple pass analysis is distinguished from an "ordinary" quadruple-pass analysis because, in an ordinary analysis, the trials on all four passes are identical. Here, only some of the four passes have trials with identical stimuli (e.g. the flattened stimuli were similar but not identical to the stimuli with natural depth variation). The quasi-quadruple pass analysis allows the three distinct decision variable correlations to take on different values. An ordinary analysis does not allow this flexibility.

For a quadruple-pass (whether quasi or not), the likelihood function takes the form $\mathcal{L}_n(\boldsymbol{\theta}) = \sum_j N_j \, \log \mathcal{L}_n^j(\boldsymbol{\theta})$ from Eq (12) above, but across sixteen response patterns

$$
j \in \left\{
\begin{array}{lllll}
[+ + ++], & & & & \\
[+ + +-], & [+ + -+], & [+ - ++], & [- + ++], & \\
[+ + --], & [+ - +-], & [+ - -+], & [- + +-], & [- + -+], \quad [- - ++], \\
[- - -+], & [- - +-], & [- + --], & [+ - --], & \\
[- - --] & & & &
\end{array}
\right\}.
$$

The individual likelihoods for these response patterns are extended from Eq (15), such that

$$
\mathcal{L}_4^j(\boldsymbol{\theta}) = \int_{s^j(c_1^z, c_2^z, c_3^z, c_4^z)} \mathcal{N}\left(\mathbf{x}^z; \mathbf{m}^z, \Sigma^z\right) \, d\mu(\mathbf{x}^z), \tag{25}
$$

with integration limits $s^j$ as described in the text proceeding Eq (13). As an example, the likelihood that a particular decision variable distribution (and set of criteria) gave rise to responses that agreed on all four passes is given by

$$
\mathcal{L}_4^{++++}(\boldsymbol{\theta}) = \int_{c_1^z}^{\infty} \int_{c_2^z}^{\infty} \int_{c_3^z}^{\infty} \int_{c_4^z}^{\infty} \mathcal{N}\left(\mathbf{x}^z; \mathbf{m}^z, \Sigma^z\right) \, d\mu(x_1^z, x_2^z, x_3^z, x_4^z). \tag{26}
$$

(We note that builtin MATLAB routines for computing an arbitrary integral of a four-dimensional normal distribution are slow and unreliable. Our code release made use of publicly available MATLAB code written for this purpose [21].)

Just as with the double-pass analysis described above, it is convenient to normalize the joint decision variable **D** in quadruple-pass analyses via application of a normalization matrix $M = \text{diag}(\frac{1}{\boldsymbol{\sigma}_T})$. In a quasi-quadruple-pass analysis, the vector $\boldsymbol{\sigma}_T$ of standard deviations is given by

$$
\boldsymbol{\sigma}_T = \begin{bmatrix} \sigma_{T\dagger} \\ \sqrt{\sigma_{T\dagger}\sigma_{T*}} \\ \sqrt{\sigma_{T*}\sigma_{T\dagger}} \\ \sigma_{T*} \end{bmatrix}, \tag{27}
$$

which results in correlation matrix

$$
\Sigma^z = \begin{bmatrix} 1 & \rho_{\dagger\dagger} & \rho_{\dagger*} & \rho_{\dagger*} \\ \rho_{\dagger\dagger} & 1 & \rho_{\dagger*} & \rho_{\dagger*} \\ \rho_{\dagger*} & \rho_{\dagger*} & 1 & \rho_{**} \\ \rho_{\dagger*} & \rho_{\dagger*} & \rho_{**} & 1 \end{bmatrix}. \tag{28}
$$

With estimates i) of the total variance of the decision variables from the two double-pass experiments (i.e. $\sigma_{T*}^2$ and $\sigma_{T\dagger}^2$) which are obtained from the thresholds, and ii) of the three decision-variable correlations between passes within and across the two double-pass experiments (i.e. $\rho_{\dagger\dagger}$, $\rho_{**}$, and $\rho_{\dagger*}$), the values of the five unknown parameters can be determined.

Estimates of decision-variable correlation are obtained by maximizing the likelihood of the data under the model. Specifically,

$$\hat{\rho}_{\dagger\dagger},\ \hat{\rho}_{\dagger*},\ \hat{\rho}_{**} = \arg\max_{\rho_{\dagger\dagger},\,\rho_{\dagger*},\,\rho_{**}} \sum_j N_j\ \log\mathcal{L}_4^j(\rho_{\dagger\dagger},\ \rho_{\dagger*},\ \rho_{**}). \tag{29}$$

We show in the next section how to solve for the contributions of the two distinct natural stimulus-driven factors—i.e. $L$ and $B$—to the variance of the decision variable.

**Determining the variability of the stimulus-driven components** We modeled natural-stimulus variability as being due to two distinct factors: luminance-pattern variability $L$ and local-depth-variability $B$ (see Eq. (22)). To obtain maximum-likelihood estimates of the variance of the luminance-pattern-driven component of the decision variable $\hat{\sigma}_L^2$, the variance of the local-depth-driven component $\hat{\sigma}_B^2$, and the covariance between these two components $\hat{\mathrm{cov}}[L,B]$, from the maximum-likelihood estimates of the three decision-variable correlations (see Eq (29)), we rearranged Eqs (23) and (24) for the variables in question. Specifically,

$$\hat{\sigma}_L^2 = \hat{\rho}_{\dagger\dagger}\hat{\sigma}_{T_\dagger}^2, \tag{30}$$

$$\hat{\mathrm{cov}}[L,B] = \hat{\rho}_{\dagger*}\hat{\sigma}_{T_\dagger}\hat{\sigma}_{T_*} - \hat{\sigma}_L^2, \tag{31}$$

$$\hat{\sigma}_B^2 = \hat{\rho}_{**}\hat{\sigma}_{T_*}^2 - \hat{\sigma}_L^2 - 2\hat{\mathrm{cov}}[L,B]. \tag{32}$$

The maximum likelihood estimates indicated in Eqs (30) and (32) are plotted in the main text (see subsection "Partitioning the variability of the decision variable" of the Results section below). The maximum-likelihood estimate of the covariance between the two components (31) tended towards zero, and can safely be ignored.

**Fitting constraints** Model parameters were fit via the quasi-quadruple-pass analysis under a pair of constraints. The first constraint was that the disparity-discrimination thresholds used in normalization matrix M (see Eq (27)) were set to values obtained from linearly constrained threshold fits (see Eq (7)). The second constraint was that the scaled covariance (i.e. correlation) between the luminance-driven and local-depth-driven components of the decision variable was constrained to lie between -1 and 1. In particular,

$$-1 < \frac{\mathrm{cov}[L,B]}{\sigma_L\sigma_B} < 1, \tag{33}$$

where the scaling factor is given by the product of the standard deviations of the two stimulus-driven components. Given that most estimates of the interaction term were near zero, we re-fit the model with the more stringent constraint that this interaction term equaled zero. Eqs (23) and (24) make clear that setting the interaction term equal to zero forces the different decision-variable correlations to have more constrained values with respect to one another than they would be constrained to have otherwise. The log-likelihoods of the models with their best-fit parameters were essentially identical, regardless of whether the interaction term was set equal to zero or not. Non-zero values of the interaction term thus carried little explanatory value.

## Between-observers decision-variable correlation

To derive an expression for between-observers decision-variable correlation, the stimulus-driven component of the decision variable is assumed to be the sum of two independent random variables. (Note that this expansion of the stimulus-driven component is not inconsistent

with the expansion used in Eq (21) above.) One is a stimulus-driven component that is shared across observers, the other is a stimulus-driven component that is private to each observer. Specifically,

$$
\begin{aligned}
D_1 &= \overbrace{(S_1 + P_1)}^{V_1} + W_1, \\
D_2 &= \underbrace{(S_2 + P_2)}_{V_2} + W_2,
\end{aligned}
\tag{34}
$$

where $S_1$ and $S_2$ are stimulus-driven components that are identically driven by the stimulus across observers (i.e. the components are proportional $S_1 \propto S_2$, or identical up to a scale factor $S_1 = aS_2$), $P_1$ and $P_2$ are the stimulus-driven components that are private to (i.e. uncorrelated between) each observer, and $W_1$ and $W_2$ are the respective noise-driven components (see Eq (9)). The total variance of the stimulus-driven component of the decision variable $V_i$ in each subject $i$ is given by $\sigma_{Ei}^2 = \sigma_{Si}^2 + \sigma_{Pi}^2$, the sum of the variances in the shared and private components. (Note the overloaded subscripts notation. Here, $i$ subscripts denote different subjects. Earlier, $i$ subscripts denoted different passes through the experiment.) Between-subjects decision-variable correlation is given by

$$
\rho_{12} = \frac{\mathrm{cov}[S_1, S_2]}{\sqrt{\sigma_{T1}^2 \sigma_{T2}^2}},
\tag{35}
$$

where $\sigma_{T1}^2$ and $\sigma_{T2}^2$ are the total variance of the decision variables in each observer. Of course, these variances include the effects of internal noise. To eliminate the impact of internal noise in the two observers, one can divide through by the square-roots of the within-observer decision-variable correlations to obtain the partial correlation

$$
\rho_{12 \cdot W} = \frac{\rho_{12}}{\sqrt{\rho_{11} \rho_{22}}} = \frac{\mathrm{cov}[S_1, S_2]}{\sqrt{\sigma_{E1}^2 \sigma_{E2}^2}},
\tag{36}
$$

where $\sigma_{E1}^2$ and $\sigma_{E2}^2$ are the variances of the stimulus-driven component of the decision variable for each observer, and $\rho_{11}$ and $\rho_{22}$ are the within-observer decision-variable correlations for each observer. This partial correlation $\rho_{12 \cdot W}$ specifies the degree to which the stimulus-driven components in two different observers are correlated with each other. High levels of this partial correlation indicate that stimulus-driven components of the two observer are highly similar.

**Estimating between-observers correlations** Estimation of between-observers decision-variable correlation within a given experiment also used the quasi-quadruple pass analysis introduced above, with a few small but important differences. The vector of standard deviations that determined the normalizing matrix is given by

$$
\boldsymbol{\sigma}_T = \begin{bmatrix} \sigma_{T_1} \\ \sqrt{\sigma_{T_1} \sigma_{T_2}} \\ \sqrt{\sigma_{T_2} \sigma_{T_1}} \\ \sigma_{T_2} \end{bmatrix},
\tag{37}
$$

where subscripts 1 and 2 indicate observer identity, rather than the experiment number. The resulting correlation matrix is given by

$$\mathbf{\Sigma}^z = \begin{bmatrix} 1 & \rho_{11} & \rho_{12} & \rho_{12} \\ \rho_{11} & 1 & \rho_{12} & \rho_{12} \\ \rho_{12} & \rho_{12} & 1 & \rho_{22} \\ \rho_{12} & \rho_{12} & \rho_{22} & 1 \end{bmatrix}. \tag{38}$$

where $\rho_{12}$ is the between-observer decision variable correlation, and $\rho_{11}$ and $\rho_{22}$ are the within-observer decision variable correlations. The maximum likelihood estimates of these parameters was given by

$$\hat{\rho}_{11},\ \hat{\rho}_{12},\ \hat{\rho}_{22} = \underset{\rho_{11},\ \rho_{12},\ \rho_{22}}{\arg\max} \sum_j N_j\ \log \mathcal{L}_4^j(\rho_{11},\ \rho_{12},\ \rho_{21}). \tag{39}$$

In each of the two experiments, all three unique pairings of observers per experiment were analyzed, so that three between-observers decision variable correlations were estimated for each condition of each experiment.

**Between-observer fitting constraints**  Constraints for quasi-quadruple between-observers analysis were similar to those for within-observer analysis. First, disparity-discrimination thresholds used in normalization matrix M (see Eq (37)) were set to values obtained from linearly constrained threshold fits (see Eq (7)). Second, the scaled partial correlation $\rho_{12\cdot W}$ between the luminance-driven and local-depth-driven components of the decision variable was constrained to lie between -1 and 1. In particular,

$$-1 < \frac{\mathrm{cov}[\mathrm{S1,S2}]}{\sigma_{E1}\sigma_{E2}} < 1. \tag{40}$$

## Spatial integration

Throughout the article, we defined the disparity of the patch to be the disparity associated with the central pixel. But there is no guarantee that human observers base their responses on the central pixel alone. It is possible–perhaps, likely–that observers based their responses on the average disparity within some spatial-integration region.

We examined whether the decision-variable correlations that we observed might be due, at least in part, to observers basing their responses on the average disparity within a fixed area, rather than on the disparity at the central pixel. We computed a new decision variable, trial-by-trial, for each of several spatial-integration areas and tested whether it provides an improved ability to account for the decision-variable correlations. We computed each new decision variable as

$$D_\mathrm{a} = \frac{\sum\limits_{\mathbf{x}} \left( \delta_\mathrm{cmp}(\mathbf{x}) - \delta_\mathrm{std}(\mathbf{x}) \right) \mathbf{w}_i(\mathbf{x})}{\sum\limits_{\mathbf{x}} \mathbf{w}_i(\mathbf{x})}, \tag{41}$$

where the window $\mathbf{w}$ defines the area of spatial integration. We computed alternative decision variables for pillbox-shaped windows having diameters ranging from a 7.5 arcmin diameter up to a 1 degree (4 to 32 pixel) diameter. The sign of the decision variable predicts the binary response. The ability of each newly computed decision variable to predict the human responses was then assessed via logistic regression.

To setup the logistic regression model, the decision variables were used as the regressor for the human binary responses. For each window-size, a single random effects model was used, conditioned by disparity pedestal, disparity-contrast, and observer. The coefficient of determination ($R^2$) was used to assess explanatory power of a given window size, and the Akaike information criterion (AIC) was used to compare models and their significance.

We also evaluated three alternative models to determine whether they could better account for the data. These models differed from the spatial-integration model described above by their decision variables, which were defined as

$$D_a = \tilde{\delta}_{cmp} - \tilde{\delta}_{std},\tag{42}$$

where $\tilde{\delta}_{cmp}$ and $\tilde{\delta}_{std}$ represent heuristics of disparity for the comparison and standard stimulus patch respectively. Each model used a different heuristic: the largest near disparity, the largest far disparity, or the maximally deviant disparity in each patch.

## Results

Three observers collected 20,000 trials each across two double-pass experiments, with the aim of determining how different types of natural stimulus variability—namely, variation in luminance-contrast patterns and variation in local-depth variation—limit sensory-perceptual performance in a depth-from-disparity discrimination task. Comparing performance between two appropriately designed double-pass experiments enables one to dissect the limits placed on performance by distinct types of stimulus-driven uncertainty versus the limits imposed by noise.

In each of the two double-pass experiments, psychometric data were collected with stimuli sampled from scene locations with two different levels of local-depth variability. There were ten conditions total in each experiment—five fixation disparities $\delta_{std}$ (i.e. disparity pedestals) at $[-11.25, -9.38, -7.5, -5.63, -3.75]$ arcmin crossed with the two levels of local-depth variability $\delta_C$ (i.e. disparity-contrast; see Methods) that ranged between $[0.025—0.117, 0.393—1.375]$ arcmin. In the first double-pass experiment, all stimuli contained natural luminance-pattern variation and natural local-depth variation. In the second double-pass experiment, "flattened" versions of the stimuli from the first experiment were used such that local-depth variation was eliminated while leaving luminance patterns essentially unaffected (see Fig 1B and 1C).

To obtain the stimuli for the experiments, thousands of stereo-image patches were sampled from a published dataset of stereo-photographs of the natural environment with co-registered laser-based distance measurements at each pixel [17]. Corresponding points were calculated directly from the distance data; a subset of corresponding points is shown in one example stereo-image (Fig 1D). Stereo-image patches with zero disparity were sampled such that the central pixels of each half-image were corresponding points associated with a given scene point along a virtual cyclopean line of sight (Fig 1E). Stereo-image patches with non-zero disparity were obtained by introducing the required amount of uncrossed disparity at the central pixel [18]. To quantify local-depth variability (i.e. disparity-contrast), ground-truth disparities were computed at each pixel directly from the distance measurements. The routines upon which the sampling procedures were built were accurate to within $\pm 5$ arcsec [18]. Hence, sampling errors are smaller than human stereo-detection thresholds for all but the very most sensitive conditions [20,22].

Stimuli were presented using a two-interval, two-alternative forced choice (2AFC) design (Fig 2A). The task was to indicate, with a key-press, which of two natural stereo-image patches

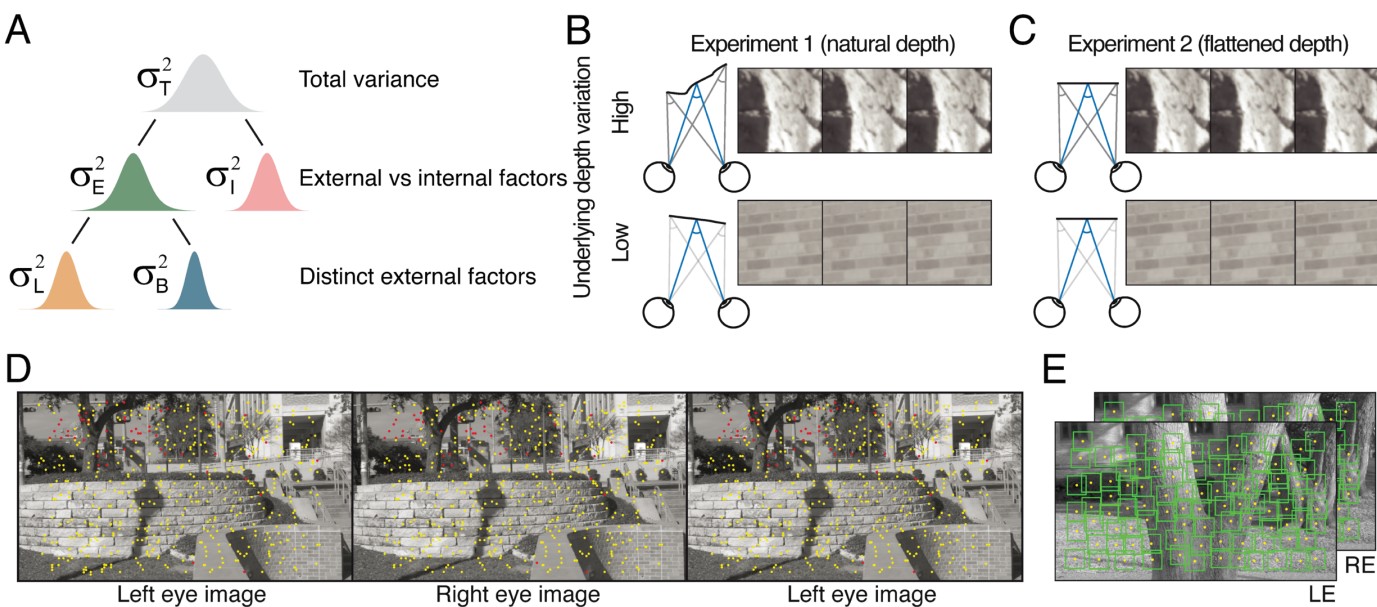

**Fig 1. Sources of uncertainty in stereo-depth perception, stereo-image database, and experimental stimuli.** (A) The total variance of the decision variable—the variable that signal-detection-theory posits that perceptual decisions are made on the basis of—is contributed to by at least two distinct sources of uncertainty: external (e.g. stimulus-driven) variability $\sigma_E^2$ and internal noise $\sigma_I^2$. The stimulus-driven component can be decomposed into distinct external factors: here, luminance-driven variability $\sigma_L^2$ and local-depth-driven variability $\sigma_B^2$. In natural viewing, luminance-driven variability depends on how luminance-contrast patterns vary across natural stimuli, and depth-driven variability depends on how local-depth structure varies across natural scenes (see B and C). Traditional psychophysical methods can determine the total variance of the decision variable. Double-pass experiments can partition the total variance into externally- and internally- driven components. The new approach used here can further partition the externally-driven component into distinct external factors. Two double-pass disparity-discrimination experiments were conducted. Both used images from hundreds of the same natural scene locations. (B) Experiment 1 used stimuli with natural-depth profiles, as quantified by disparity-contrast (see Methods Eq (1)), was either high (*top row*) or low (*bottom row*). (C) Experiment 2 used the same stimuli but with flattened versions of the natural-depth profiles. The flattened stimuli had the same luminance profiles as the stimuli in Experiment 1, but had no local-depth variation. (D) Example natural stereo-image from which natural stimuli were sampled for the experiments, taken from a publicly available database [17], licensed under CC BY-NC-ND 4.0. Corresponding points, overlaid in yellow, were calculated directly from laser-range-based ground-truth distance data at each pixel. Points in one image without a valid corresponding point in the other (e.g. half-occluded scene regions) are colored red. Divergently-fuse the left two images, or cross-fuse the right two images, to see the scene in stereo-3D. (E) Another example natural stereo-image with patches that were vetted for inclusion in the experimental stimulus set (*boxes*; see Methods). For purposes of visualization, depicted patches are four times wider (4×4°) than those used in the actual experiments (1×1°).

appeared to be farther behind the screen. On each trial, we assume that disparity estimates are obtained for the standard and comparison stimuli: $\delta_{std}$ and $\delta_{cmp}$, respectively. Each of these estimates is affected both by properties of the standard and comparison stimuli, and by noise. The decision variable is then obtained by subtracting the standard disparity-estimate from the comparison disparity estimates. Distributions of these disparity-estimate and decision-variable distributions are shown in Fig 3B.

## Decision-variable correlation

The decision variable underlying performance is given by

$$D = V + W, \tag{43}$$

where $V$ is captures the effect of externally-driven, stimulus-based variability and $W$ is a sample of internal noise.

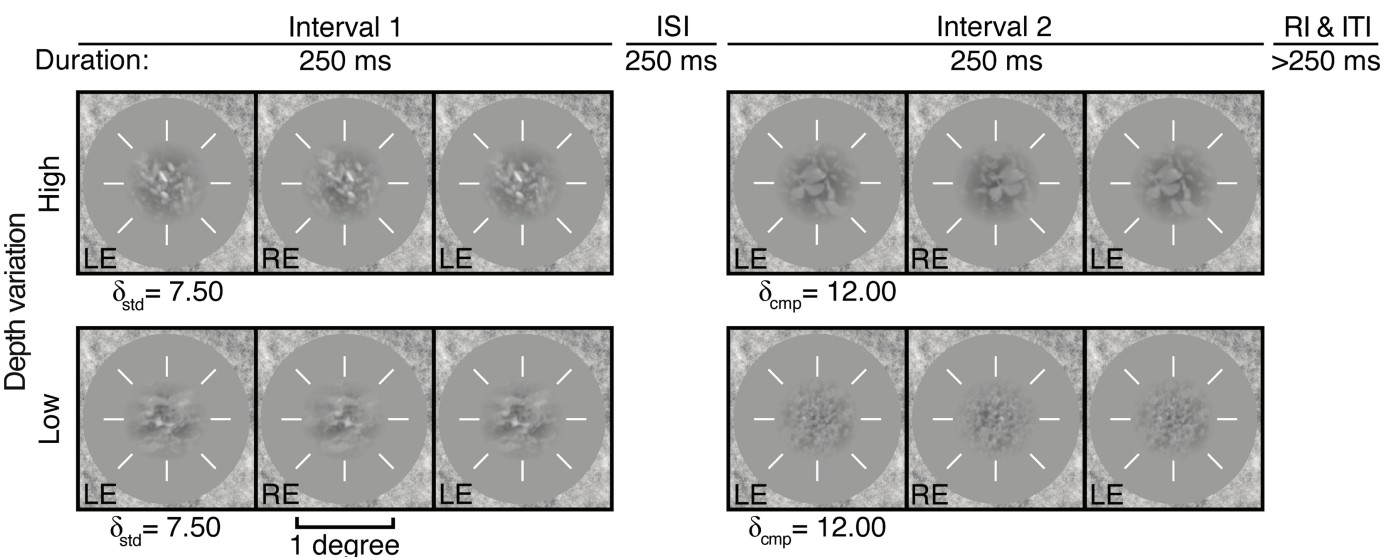

**Fig 2. Example trials.** Stimuli were presented using a two-interval, two-alternative forced choice (2AFC) design. The task on each trial was to indicate which of two briefly presented (250 ms) stereo-defined stimuli appeared to be farther behind the display. Example trials from high (*top*) and low (*bottom*) disparity-contrast conditions of experiment 1. Over the course of a single pass, a unique natural image patch was presented for every trial and interval with binocular-disparity-defined depth at each patch-center. Each patch either had a standard or comparison disparity value ($\delta_{std}$ and $\delta_{cmp}$, respectively). The interval in which the standard (and comparison) appeared was randomized. Each natural image patch was unique across all trials and intervals. Patches were 1 degree in diameter and presented at the center of a mean-luminance gray area with a fixation crosshairs, surrounded by a 1/f noise field.

The double-pass experimental design (Fig 3A), like a typical (single-pass) experimental design, allows one to estimate the variance of the decision variable. Assuming conditional independence between externally- and internally-driven components, the total variance of the decision variable is given by

$$\sigma_T^2 = \sigma_E^2 + \sigma_I^2, \tag{44}$$

where $\sigma_E^2$ is the variance of the externally-driven component and $\sigma_I^2$ is the variance of the internally-driven component. The total variance of the decision variable can be computed directly from the discrimination threshold (Fig 3B-C). Specifically, for a certain definition of threshold-level performance which we used here (i.e. $d' = 1.0$), the total variance of the decision variable simply equals the square of the discrimination threshold (i.e. $\sigma_T^2 = T^2$; see Methods Eq (5)).

The double-pass experimental design, more uniquely, allows one to estimate decision-variable correlation (Fig 3D-F). Decision-variable correlation indicates the degree to which the trial-by-trial values of the decision variable are correlated across passes. It is given by

$$\rho = \frac{\sigma_E^2}{\sigma_T^2}, \tag{45}$$

being equal to the proportion of total variability in the decision variable that is due to factors that are common across repeated presentations of the same trial (e.g. external stimulus variability) versus those that are not (e.g. internal noise). Decision-variable correlation is estimated from the repeatability of observer responses across passes (Fig 3D-E; see below). On

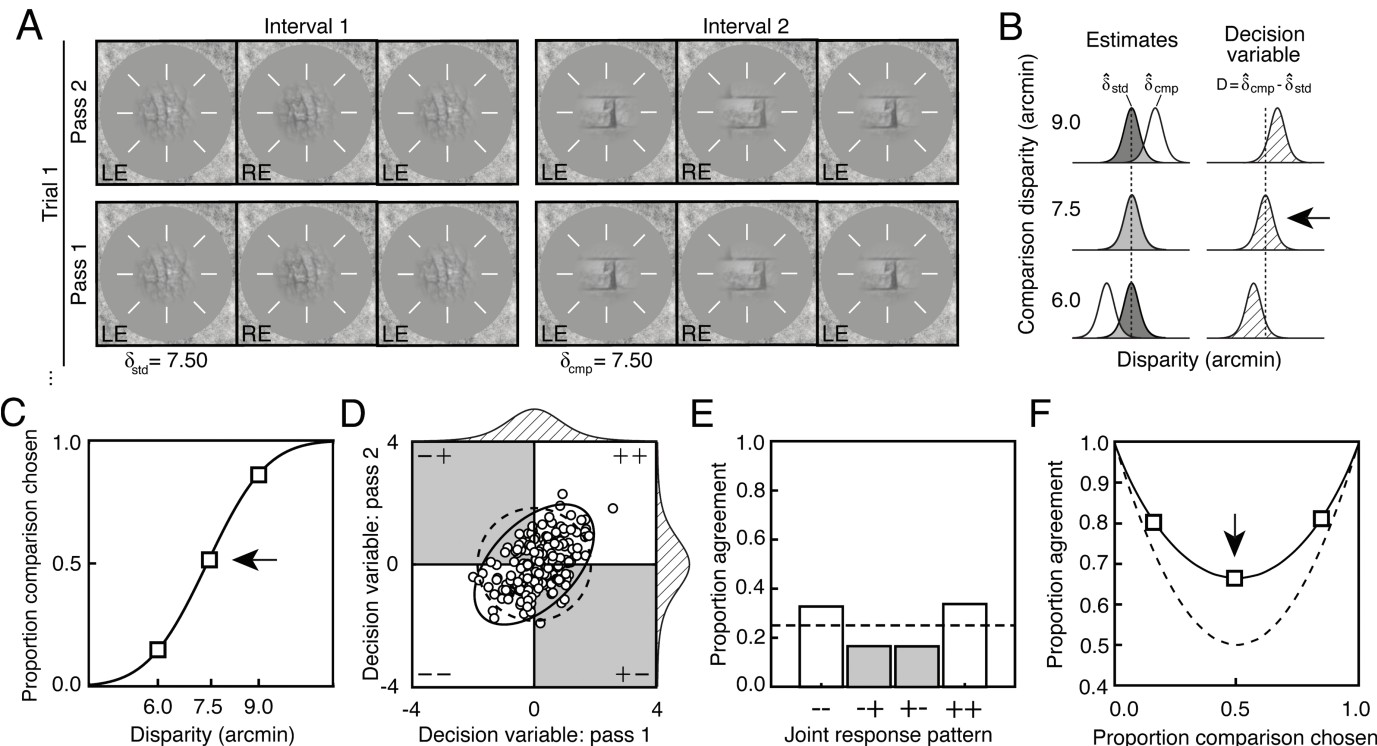

**Fig 3. Double-pass experimental design.** (A) Each pass of a double-pass experiment is composed of a large number of unique trials, presented one time each. Trials are identical between passes. (B) Standard and comparison disparity estimate distributions for each of three comparison disparity levels (*left*) and corresponding decision variable distributions (*right*). Each decision variable distribution is obtained simply by subtracting the standard disparity estimate from the comparison disparity estimate on each trial. (C) Psychometric data for stereo-depth discrimination with fitted cumulative Gaussian curve, collapsed across both passes of a double-pass experiment. Threshold or standard deviation of the decision variable is estimated from the variance parameter of the curve. Psychometric data are binary, indicating whether the comparison stimulus was chosen (+) or not (−). Different decision-variable distributions (B) underlie performance at each point on the psychometric function. Data at 0.5 proportion comparison chosen (*arrow*) are the least informative for estimating discrimination thresholds, but the most informative for estimating decision variable correlation (see *D* and *E*). (D) Distribution of joint decision variable (*ellipses*) and samples (*dots*) across both passes of a double-pass experiment. Samples in each of the four different quadrants will yield one of four possible joint responses across passes (−−, −+, +−, ++), two of which indicate agreement (++ and −−). Decision-variable correlations larger than 0.0 evince shared sources of response variability across passes. Dashed ellipse shows joint decision-variable distribution if observer responded completely by chance (correlation of zero). (E) Histograms show the expected proportion of each of the four joint response types for the joint-decision-variable distribution shown in *B*. (F) Proportion of between-pass agreement as a function of proportion comparison chosen. Solid line shows best fit to the data. Dashed line shows expected agreement if the observer responded completely by chance (correlation of zero).

each trial of each pass, the observer reports either that the comparison stimulus appeared farther away than the standard stimulus (+), or that the comparison stimulus appears closer than the standard stimulus (−). Upon completion of both passes, the observer will have made a particular joint response on each unique trial, out of four possible joint responses (−−,−+, +−, ++). When decision-variable correlation equals zero—as it will be when internal noise is the only source of variability in the decision variable—response agreement is expected to be at chance levels (see Fig 3D-F, dashed lines). When decision-variable correlation is high—as it will be when external factors (e.g. nuisance stimulus variability) are the dominant source of variance in the decision variable—response agreement will be high.

Decision-variable correlation, like other important quantities in signal detection theory (e.g. $d'$), must be estimated from a set of binary observer responses (Fig 3D-F). We computed how repeatable observers' responses were (i.e. how often observer responses agreed) across

the repeated presentations of the same stimuli in the first and second passes of the double-pass experiment (see Fig 3 and Methods). From the level of response agreement in a given condition, we used maximum-likelihood techniques to estimate decision-variable correlation across passes.

Decision-variable correlations reflect the *relative* contributions of each individual source of variability in the decision variable (Eq (43)). A change in decision-variable correlation between conditions could result from an increase in one source of variability, a decrease in the other, or a combination of both. Discrimination thresholds provide an *absolute* measure of the total variance in the decision variable, but they do not indicate the relative contribution of external (e.g. stimulus-driven) and internal (e.g. noise-driven) sources of variability (Eq (44)). Together, discrimination thresholds and decision-variable correlation can be used to determine the absolute contribution of stimulus-driven and internal-noise-driven sources of variability to the decision variable (see Eqs (10) and (18)). From estimates of decision-variable correlation (Eq (45)) and of the total variance of the decision variable (Eq (44)), the variances of the externally- and internally-driven components of the decision variable can be computed (see Methods, and below).

### Experiment 1: Natural stimuli with natural-depth profiles

Fig 4 shows raw data from one individual observer in the first double-pass experiment which used stimuli having natural-luminance and natural-depth profiles. Psychometric data and function fits showing proportion comparison chosen are presented in Fig 4A. The slopes of the psychometric functions decrease systematically both as disparity-contrast increases from low to high (top vs. bottom), and as disparity pedestal increases (psychometric functions, left to right). These patterns show that, as the surfaces to be discriminated become more non-uniform in depth (i.e. have higher disparity-contrast), and as they move farther from the fixated distance, discrimination thresholds increase.

Response agreement data and fits for the same observer are shown in Fig 4B. The corresponding estimates of decision-variable correlation in each condition are indicated at the top of each subplot. In all conditions, response agreement is systematically higher than expected under the assumption that decision-variable correlation equals 0.0. Indeed, decision-variable correlation is approximately equal to 0.5, on average across the conditions. Thus, the relative contributions of externally- and internally-driven components to the variance of the decision variable are similar (i.e. $\sigma_E^2 \approx \sigma_I^2$; see Eqs (44) and (45)). External and internal sources limit performance near-equally. Further, decision-variable correlation is always higher in the high than in the low disparity-contrast conditions (see the inset values of $\rho$ in each subplot). The increase in decision-variable correlation with the level of disparity-contrast entails that the threshold increases are due to more substantial increases in the variance of the stimulus-driven than of the noise-driven component of the decision variable.

Fig 5A shows how stereo-based depth discrimination thresholds change with fixation error (i.e. disparity pedestal) and local-depth variability (i.e. disparity-contrast) for each individual observer, and the observer average. For both disparity-contrast conditions, discrimination thresholds are well-characterized by an exponential function, the signature of which is a straight line on a semi-log plot. This exponential rise in discrimination threshold with pedestal disparity is a classic empirical finding [20,22–25], and is predicted by a normative image-computable model of optimal disparity estimation with natural stereo-images [8]. The current result provides a psychophysical demonstration that the classic exponential law of human disparity discrimination generalizes to natural stimuli. Because this pattern is robust

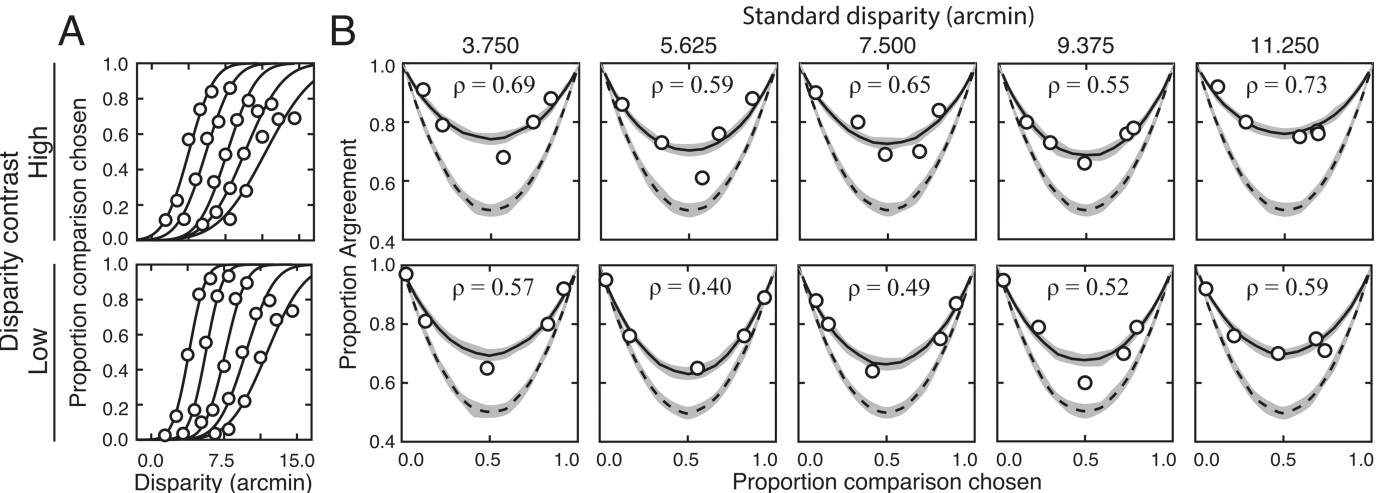

**Fig 4. Discrimination thresholds, response agreement, and estimates of decision-variable correlation results for one observer.** (A) Response data (*points*) and psychometric curves for each condition. Thresholds increase systematically with disparity pedestal and with disparity-contrast. (B) Human response agreement and fitted agreement curves for each condition. Thresholds and decision-variable correlation was used to determine relative impact between sources of performance variability. Dashed lines shows expected agreement if the observer responded completely by chance (correlation of zero).

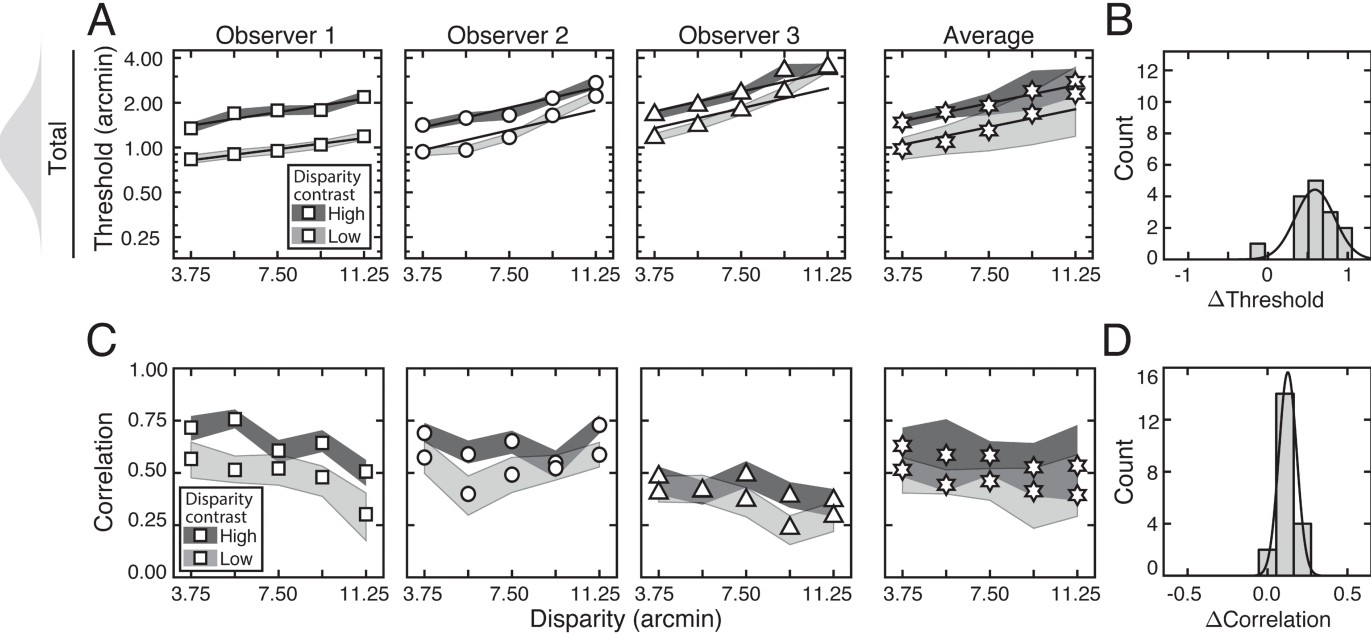

**Fig 5. Experiment 1 discrimination thresholds and decision-variable correlations.** Stimuli in Experiment 1 had naturally varying local-depth variation. (A) Discrimination thresholds as a function of disparity pedestals, for different disparity-contrast levels (*shades*), for each observer and the observer average (*columns*). For individual observers, shaded regions indicate 68% confidence intervals for each condition, generated from 10,000 bootstrapped datasets. For the observer average, shaded regions indicate across-observer standard deviations. Lines represent exponential fits to the data in each disparity-contrast condition (see Methods). Discrimination thresholds are equal to the square-root of the total variance of the decision variable (see Eq (5)). (B) Histogram of threshold differences in the high and low disparity-contrast conditions, collapsed across disparity pedestal and individual observers. Curves indicate best-fit normal distributions to the data. (C) Estimated decision-variable correlation in the same conditions for each observer and the observer average. (D) Histogram of differences in decision-variable-correlation differences between the high and low disparity-contrast conditions, collapsed across disparity pedestal and individual observers.

to the particular stimuli that are used to probe performance, it should be thought of as a feature of how the visual system processes disparity, rather than a consequence of the particular stimuli used to probe performance.

Discrimination thresholds are also higher for stimuli with high disparity-contrast than they are for stimuli with low disparity-contrast. Hence, local-depth variability harms depth discrimination performance. As disparity-contrast increases, thresholds shift vertically in the semi-log plots, such that the two sets of thresholds are parallel to one another, indicating that the threshold increases with disparity-contrast are multiplicative. Fig 5B visualizes these threshold differences as a histogram, collapsed across all disparity pedestals and observers. Clearly, the histogram of threshold differences is substantially shifted to the right of zero, which confirms that thresholds increase with disparity constraint.

The fact that disparity-contrast degrades discrimination performance should surprise no one [26–29]. Increased local-depth variability entails that the left- and right-eye images have more local differences between them. These more pronounced local differences make the stereo-correspondence problem more difficult to solve. The increased difficulty in solving the correspondence problem should, in turn, make stereo-based depth discrimination more difficult. This increase in difficulty is what we observe in our results. However, as we will see, this expected degradation in discrimination performance with disparity-contrast is partly due to a surprising underlying cause (see below).

Decision-variable correlations in each condition for each observer, and for the observer average are shown in Fig 5C. In each and every condition, decision-variable correlation is higher in the high disparity-contrast condition than in the low disparity-contrast condition (Fig 5D). This pattern of results indicates that as disparity-contrast increases and the task becomes harder, there is an increase in the proportional impact of external, stimulus-driven components on the decision variable—that is, observer responses become more repeatable, not less.

The externally- and internally-driven contributions to threshold were computed from the estimates of decision-variable correlation and the total variance of the decision variables (i.e. discrimination-thresholds squared (see Eqs (18) and (19)), and are shown in Fig 6. As with the discrimination thresholds (see Fig 5A)—which reflect the total variance of the decision variable—these individual components also tend to increase exponentially with disparity pedestal (i.e. linearly on semi-log axes; see Fig 6A). However, disparity-contrast impacts these two components differently. The variance of the externally-driven component scales with disparity-contrast (Fig 6A top row), and substantially so, whereas the variance of the internally-driven component changes more modestly (Fig 6A bottom row). Thus, the increase in discrimination thresholds with disparity-contrast can be attributed primarily to increases in the variance of the externally-driven (i.e. stimulus-driven) component of the decision variable. The histograms in Fig 6B emphasize this point. They show histograms of the difference in variance between the high and low disparity-contrast conditions in each component, across all observers and disparity pedestals. Clearly, the effect of disparity-contrast on the externally-driven component is more pronounced than the effect on the internally-driven component.

As noted, the fact that discrimination thresholds increase with local-depth variability is to be expected [26]. What is unexpected is that a substantial portion of the threshold increases are attributable to factors that make responses more repeatable on successive presentations of the same stimulus. The implication is that, in natural scenes, local-depth variability does not simply make disparity-based depth discrimination noisier, as might be expected if local-depth variability simply made the binocular matching process more unreliable. Rather, the

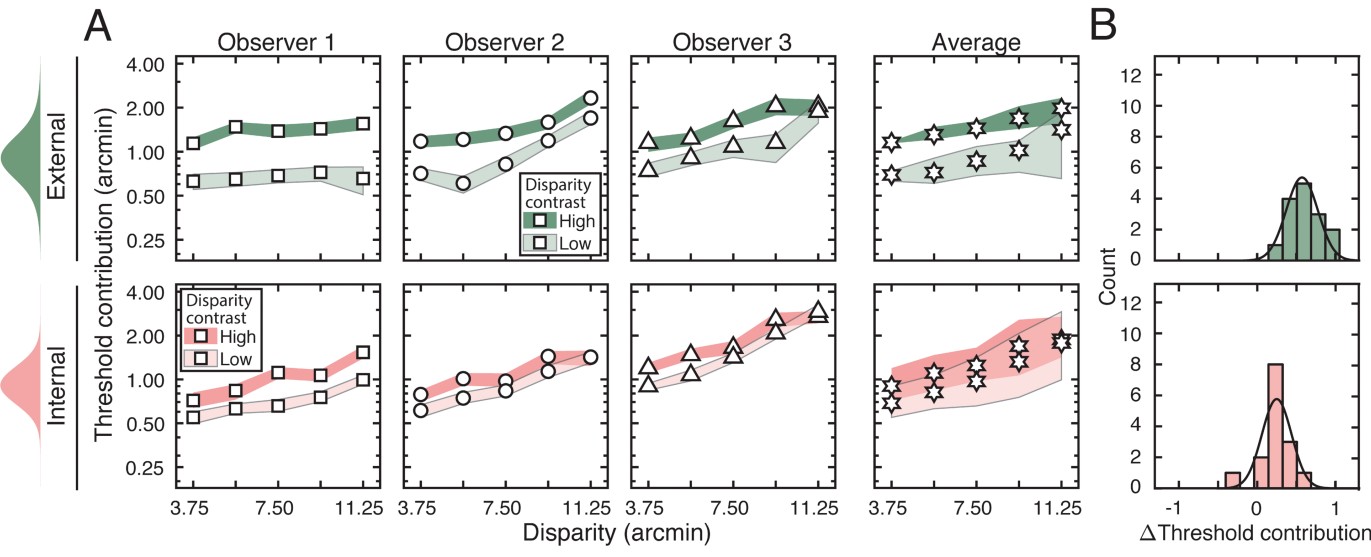

**Fig 6. External stimulus-driven and internal noise-driven contributions to thresholds in Experiment 1.** Estimated external stimulus-driven (*top row*) and internal stimulus-driven (*bottom row*) contributions to threshold, at all disparity and disparity-contrast conditions, for each observer and the observer average. For observers, bounds of shaded regions indicate 68% confidence intervals for each condition, generated from 10,000 bootstrapped samples. For the observer average, bounds indicate standard deviations. Threshold contribution reflects the variances $\sigma_E^2$ and $\sigma_I^2$ of the stimulus-driven and internal- noise-driven components of the decision variable, respectively (see Methods). B. Histograms of differences between high and low disparity-contrast conditions for both externally- and internally-driven components (*top row* and *bottom row* respectively).

results suggest that local-depth variability biases the observer, stimulus-by-stimulus, to perceive more or less depth in a manner that is repeatable across repeated stimulus presentations. The results therefore imply that, at least in principle, observer errors on each individual stimulus should be predictable. Developing image-computable models that enable stimulus-by-stimulus prediction of depth estimation performance in depth-varying natural scenes is an interesting direction for future work [12].

One potential source of observer repeatable error was that observers were not making disparity estimates based on the very most central pixels of each stimulus. Instead, observers could have been averaging disparities within a window of spatial integration. We investigated this possibility using logistic regression (see Methods), by asking whether disparities averaged within spatial integration windows of fixed size, across a range of sizes, could better account for the observer responses than the disparities associated with the central pixel of each patch. We found that all fixed window sizes accounted for the data equally well. Changing the size of the spatial integration window produced no improved ability to account for explainable variance (all $R^2 < 0.01$; S1 Fig A). And the Akaike information criterion (AIC) indicated that none of tested spatial integration window sizes produced a significantly better account of the data than the smallest window size that was implicitly assumed throughout the rest of the paper. We also investigated whether the largest near disparity, largest far disparity, and maximally deviant disparity of each patch could account for differences in performance. These analyses yielded similar results (all $R^2 < 0.01$; S1 Fig B). None of these models produced a better account of the data. The primary results should be considered representative.

Another way to investigate the degree to which stimulus-based variability is predictable is to examine between-observer performance similarities. We assessed whether between-observer-threshold variability is more attributable to differences in the effect of external factors (e.g. stimulus-based variability) or internal factors (e.g. noise) across observers. Fig 7

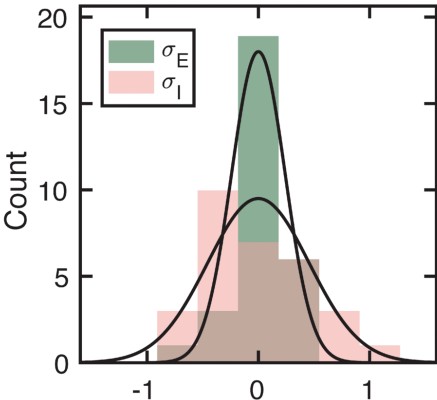

**Fig 7. Between-observer variability is primarily attributable to differences in internal noise.** Observer-mean subtracted estimates of externally-driven $\sigma_E^2$ (green) and internally-driven $\sigma_I^2$ (pink) components of the decision variable, histogrammed across conditions. Black lines represent best-fit normal distributions. Across the high and low disparity-contrast conditions, the fraction of between-observer variance explained by the internally-driven component for Experiment 1 was 0.81 ($p = 2.0 \times 10^{-4}$, $F = 0.23$ where $F$ is the test statistic of a two-sample F-test).

shows how the external, stimulus-based and internal, noise-based contributions to threshold vary across observers relative to the between-observer mean. Between-observer variation in the externally driven-component of the decision variable is substantially smaller than in the internally-driven component (Fig 7). The stimulus-driven component of the decision variable is very similar across human observers, and does not contribute substantially to between-observer differences in discrimination threshold. Because the external drive to the decision variable is consistent across observers, it implies that the stimulus-specific computations performed by the human visual system are stable across observers (also see below). Hence, between-observer variability is primarily due to differences in internal noise.

## Experiment 2: Natural stimuli with flattened depth profiles

The second double-pass experiment made use of natural stimuli having "flattened" depth profiles (see Fig 1C). The luminance profiles of these stimuli are essentially unchanged from those in the first experiment, because they were derived from the exact same scene locations, but the disparity-contrasts of all stimuli were set equal to zero. Thus, in Experiment 2, the nominal "high disparity-contrast" and "low disparity-contrast" stimuli had zero disparity-contrast, even though the luminance profiles were drawn from scene regions originally associated with high and low levels of local-depth variability.

The primary aim of the second double-pass experiment is to make it possible to partition the effects of variation in natural luminance-contrast patterns and local-depth variation in limiting stereo-depth discrimination. Doing so requires analyzing the data from both experiments simultaneously. Before turning to this joint analysis of the psychophysical data from both double-pass experiments, we first present the results of the second experiment on their own.

Fig 8 shows discrimination thresholds (i.e. the square-root of the total variance of the decision variable), and decision-variable correlations across all conditions in Experiment 2, for each individual observer and the observer average. There is one marked change in the patterns in the data as compared to the first experiment. Discrimination thresholds (Fig 8A and

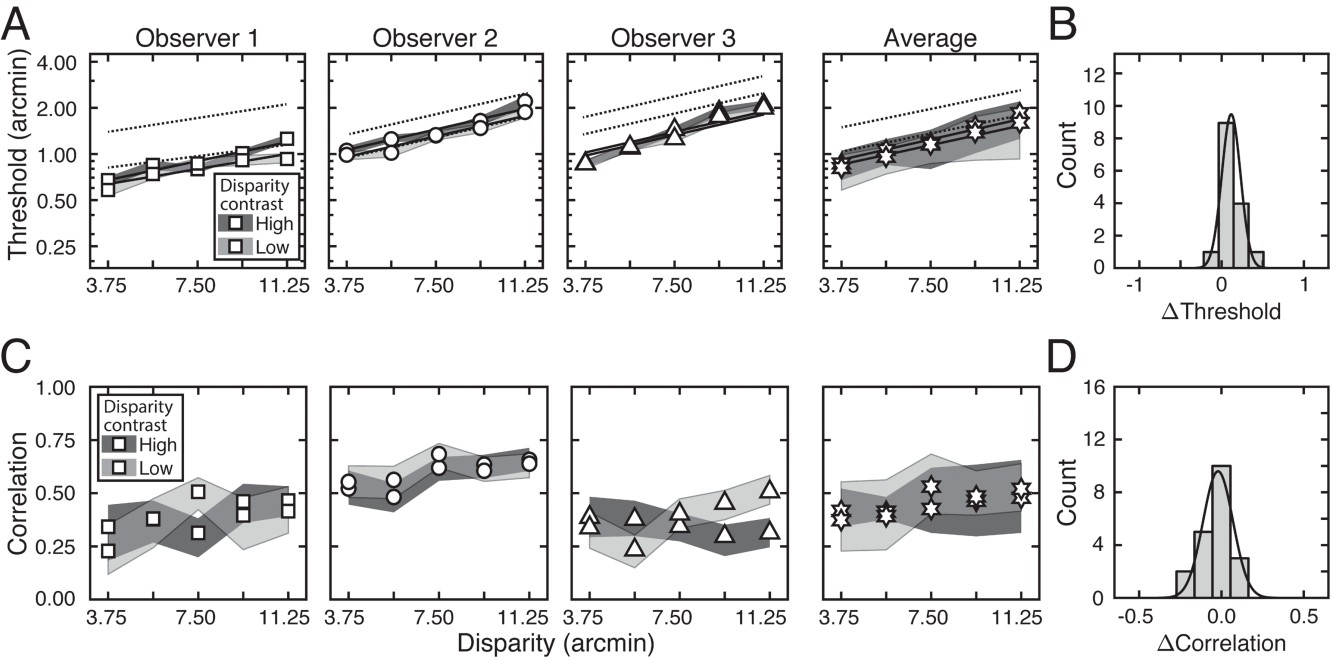

**Fig 8. Experiment 2 disparity discrimination thresholds and decision-variable correlation.** Experiment 2 stimuli were flattened (i.e. had zero local-depth variability), but otherwise had the same luminance contrast patterns as those in Experiment 1. (A) Discrimination thresholds as a function of disparity pedestal, for different disparity-contrast levels (*shades*), for each observer and the observer average (*columns*). Unlike in Experiment 1, there is little to no effect of nominal disparity-contrast on threshold. For individual observers, shaded regions indicate 68% confidence intervals for each condition, generated from 10,000 bootstrapped datasets. For the observer average, shaded regions indicate standard deviations. Solid lines represent exponential fits to the data. Dotted lines represent the exponential fits to the threshold data from Experiment 1 (see Fig 6A). (B) Histogram of threshold differences in the high and low disparity-contrast conditions, collapsed across disparity pedestal and individual observers. Curves indicate best-fit normal distributions to the data. (C) Estimated decision-variable correlation in the same conditions for each observer and the observer average. Decision-variable correlations are systematically lower than those in Experiment 1 (see Fig5). (D) Histogram of decision-variable-correlation differences in the high and low disparity-contrast conditions, collapsed across disparity pedestal and individual observers.

8B) and decision-variable correlations (Fig 8C and 8D) are now largely unaffected by nominal disparity-contrast. There are also consistent decreases in thresholds and decision-variable correlations, as compared to Experiment 1 (see Fig 5). These results imply that a source of stimulus-driven variance in the decision variable that increases response agreements across repeated stimulus presentations, has been removed from the stimuli.

Analysis of the external (stimulus-driven) and internal (noise-driven) contributions to threshold lead one to the same conclusion: flattening the stimuli removes a stimulus-driven source of variance in the decision variable that is due to local-depth variability (Fig 9). Neither the external drive to threshold (Fig 9, top row), nor the internal drive to threshold (Fig 9, bottom row), is affected by nominal disparity-contrast.

Of course, this change in the pattern of results makes sense. The "high disparity-contrast" and "low disparity-contrast" stimuli in Experiment 2 had been associated with depth varying regions of natural scenes in Experiment 1, but they were flattened for the current experiment. So the result is not unexpected. But it is also not guaranteed. The effect of natural depth variability in bumpier (higher disparity-contrast) scene regions on the decision variable could have been correlated with the effect of natural luminance-contrast patterns such that, even with flattened stimuli, the associated luminance profiles would have generated higher discrimination thresholds. That is, luminance profiles associated with scene locations having

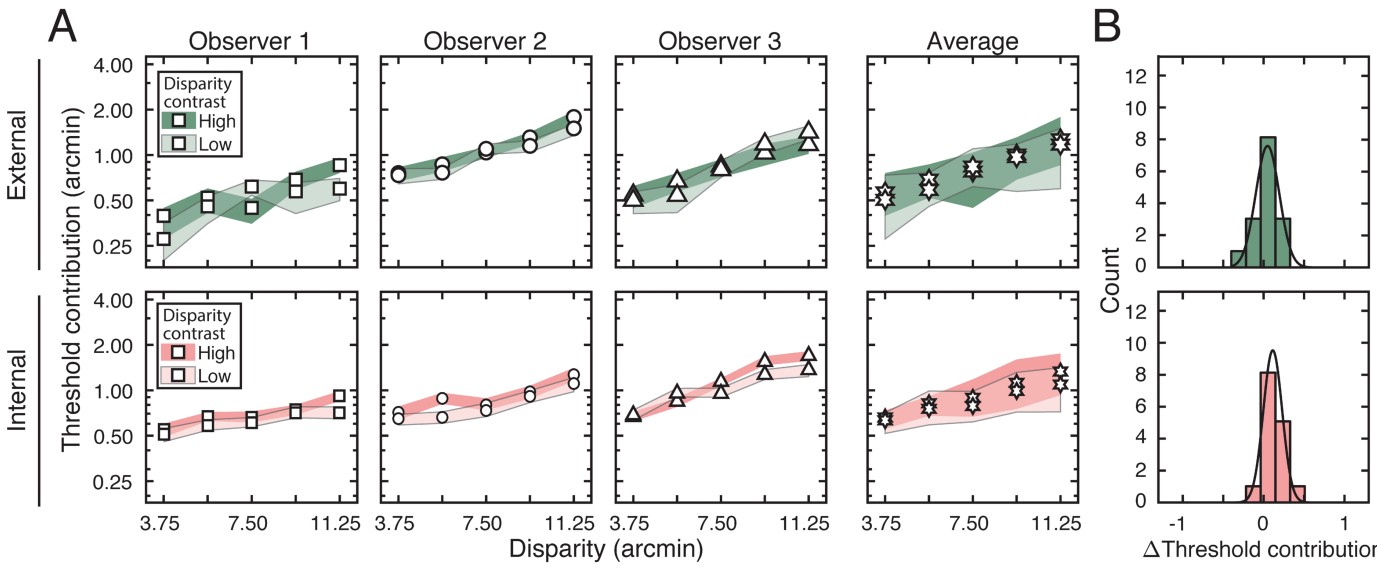

**Fig 9. External stimulus-driven and internal noise-driven contributions to thresholds in Experiment 2.** (A) Estimated external stimulus-driven (*top row*) and internal stimulus-driven (*bottom row*) contributions to threshold, at all disparity and disparity-contrast conditions, for each observer and the observer average. For observers, bounds of shaded regions indicate 68% confidence intervals for each condition, generated from 10,000 bootstrapped samples. For the observer average, bounds inidcate across-observer standard-deviations. Threshold contribution reflects the variances $\sigma_E^2$ and $\sigma_I^2$ of the stimulus-driven and internal noise-driven components of the decision variable, respectively (see Methods). Note that, in comparison to the results of Experiment 1, there is hardly any effect of disparity-contrast on the stimulus-driven contributions to threshold. (B) Histograms of differences between high and low disparity-contrast conditions for both externally- and internally-driven components (*top row* and *bottom row* respectively).

greater local-depth variability could themselves have been more difficult to discriminate, even after stimulus-flattening. The current results suggest that this is not the case.

Because of the fact that, in the first double-pass experiment, high disparity-contrast stimuli yielded high levels of externally-driven variance in the decision variable and low disparity-contrast stimuli yielded lower levels of externally-driven variance (see Fig 6A), the current results strongly imply that a stimulus-driven, and repeatable, source of variability has been removed from the decision variable. The flattened stimuli of the second double-pass experiment also yield the lowest levels of externally-driven variability. Together, these results imply that stimulus flattening removes a distinct source of variability from the decision variable. This idea is tested more rigorously below.

### Partitioning sources of variability in natural stimuli

Here, we show that stimulus-driven variability in the decision variable can be partitioned into separate factors that depend on natural-luminance and natural-depth structure. These sources of variability—natural-luminance structure and natural-depth structure—have distinct and largely separable effects on human performance. To determine the importance of these two factors, and to test whether these factors interact, we compared human performance across the four passes of the two double-pass experiments with flattened and natural stimuli. We refer to this comparative analysis as a quasi-quadruple-pass analysis (see Methods). (As noted in Methods, ordinary quadruple-pass experiments—to the extent that quadruple-pass experiments are ever ordinary—present *exactly* the same stimuli across all four passes. Our experiments presented closely related, but not identical, stimuli across the four passes of the two double-pass experiments, hence the "quasi-quadruple-pass" moniker.)

Luminance-contrast-pattern variability was essentially the same in both double-pass experiments, and was thus the same across all four passes. However, because the second double-pass experiment used flattened stimuli—which prevents local-depth variability from directly influencing the variance of the decision variable—natural-luminance variation is the only remaining stimulus factor that can contribute to the decision variable because natural-depth variability has been eliminated. The quasi-quadruple-pass analysis allows one to determine how these two factors combine and/or interact to limit performance.

To understand the reasoning behind the quasi-quadruple-pass analysis, it is useful to write out expanded expressions for the decision variable (also see Eq (43)). The expanded expression for the decision variable is given by

$$D = \overbrace{(L + B)}^{V} + W,$$

(46)

where $L$ and $B$ are luminance-profile-driven and local-depth-variability-driven contributions to the decision variable (which sum to the total stimulus-driven contribution $V$), and $W$ is a sample of internal noise.

In the double-pass experiment with natural-luminance and depth profiles (Exp. 1), the expressions for the total variance of the decision variable and for decision-variable correlation across passes, in terms of the variance of these newly articulated components (i.e. $L$ and $B$ in Eq (46)), are given by

$$\sigma_{T_*}^2 = \overbrace{\underbrace{(\sigma_L^2 + \sigma_B^2 + 2\mathrm{cov}[L, B])}_{\text{unknowns}}}^{\sigma_{E_*}^2} + \sigma_{I_*}^2,$$

(47)

$$\rho_{**} = \frac{\sigma_{E_*}^2}{\sigma_{T_*}^2} = \frac{\sigma_{E_*}^2}{\sigma_{E_*}^2 + \sigma_{I_*}^2},$$

(48)

where $\sigma_L^2$ and $\sigma_B^2$ are the variances of the components driven by luminance-pattern and local-depth variability, the interaction term $\mathrm{cov}[L, B]$ is the covariance between them (if it exists), $\sigma_{E_*}^2$ is the external (stimulus-driven) variance, and $\sigma_{I_*}^2$ is the variance of internal noise. The external stimulus-driven- and internal noise-driven variances can be solved from the equations for total variance and decision-variable correlation (Eqs (47) and (48)). But there are not enough equations to separately determine the values of the three unknown factors: the variance $\sigma_L^2$ of component driven by luminance-pattern variability, the variance $\sigma_B^2$ of component driven by local-depth variability, and the covariance $\mathrm{cov}[L, B]$ between the luminance and depth-driven components. Fortunately, the second double-pass experiment allows one of these unknown factors—the variance of the luminance-driven component of the decision variable—to be determined.

In the second double-pass experiment with natural-luminance profiles and flattened-depth profiles (Exp. 2), the expanded expression for the decision variable is given by

$$D = \overbrace{L}^{V} + W.$$

(49)

Note that the disparity-contrast-driven component $B$ that is present in the first experiment does not appear in Eq (49), because disparity-contrasts were set equal to zero when the stimuli were flattened. The corresponding expressions for the variance of the decision variable,

and decision-variable correlation, are given simply by

$$\sigma_{T\dagger}^2 = \overbrace{\sigma_L^2 + \sigma_{I\dagger}^2}^{\sigma_{E\dagger}^2},$$ (50)

$$\rho_{\dagger\dagger} = \frac{\sigma_{E\dagger}^2}{\sigma_{T\dagger}^2} = \frac{\sigma_{E\dagger}^2}{\sigma_{E\dagger}^2 + \sigma_{I\dagger}^2},$$ (51)

where, again, $\sigma_L^2$ is the luminance profile driven variance, $\sigma_{E\dagger}^2$ is the external stimulus-driven variance, and $\sigma_{I\dagger}^2$ is the internal-noise-driven variance associated with the flattened stimuli. Just as before, the external and internal variances can be estimated from Eqs (50) and (51). Now, the variance of the luminance-pattern-driven component $\sigma_L^2$ is easily obtained because it exactly equals the variance of the externally-driven component. Also note that in this experiment, because local-depth variability is absent, the variance of the disparity-contrast-driven component is zero. But there are still two remaining unknowns.

Here is where the quasi-quadruple-pass analysis proves useful. By computing decision-variable correlation across passes of the two different double-pass experiments, an additional equation is obtained. Decision-variable correlation between passes across experiments is given by

$$\rho_{\dagger*} = \frac{\sigma_L^2 + \mathrm{cov}[L, B]}{\sigma_{T\dagger}\ \sigma_{T*}}.$$ (52)

With this expression, we now have the number of equations necessary to determine the unknowns. Using maximum-likelihood techniques, we fit all three decision-variable correlations ($\hat{\rho}_{\dagger\dagger}$, $\hat{\rho}_{**}$, and $\hat{\rho}_{\dagger*}$) simultaneously from the data in both experiments with the quasi-quadruple-pass analysis (see Eq (28)), and then solved algebraically the system of equations specified by Eqs (47), (48) and (50)–(52) for the unknown parameters. This approach guarantees that shared factors between equations are consistent with one another.

Before proceeding to the main results, we briefly note that we have already estimated decision-variable correlation across passes in the first experiment and in the second experiment—$\hat{\rho}_{**}$ and $\hat{\rho}_{\dagger\dagger}$, respectively—, in each case only using data from the respective experiment in isolation. When carrying out the quasi-quadruple-pass analysis, the estimates of the within-experiment decision-variable correlations ($\hat{\rho}_{**}$ and $\hat{\rho}_{\dagger\dagger}$) and the variances of the externally- and internally-driven components ($\sigma_E^2$ and $\sigma_I^2$ ) are not guaranteed to be the same as when they are estimated with the data from only one isolated experiment (see Fig 6 and 9). Reassuringly, however, the estimates from the quasi-quadruple-pass analysis are very similar to those previously estimated. This consistency supports the claim that factors assumed to be common to both experiments are in fact common to both experiments (see Fig 10). The consistency by which these parameters vary across experiments and observers suggests that each component of the decision variable is indeed driven by the natural-image property—or a tight co-variate of the property—that is said to drive it.

Fig 11 shows the recovered values of the luminance- and depth-driven components of the decision variable—$\sigma_L^2$ and $\sigma_B^2$, respectively—and their interaction term $\mathrm{cov}[L, B]$, that were obtained from the quasi-quadruple-pass analysis (see above; also see Methods). The variances of both the luminance-driven and local-depth-driven components clearly increase with disparity pedestal for all conditions and observers. This pattern is similar to the patterns in all previous plots. More interestingly, whereas the luminance-driven component is very nearly

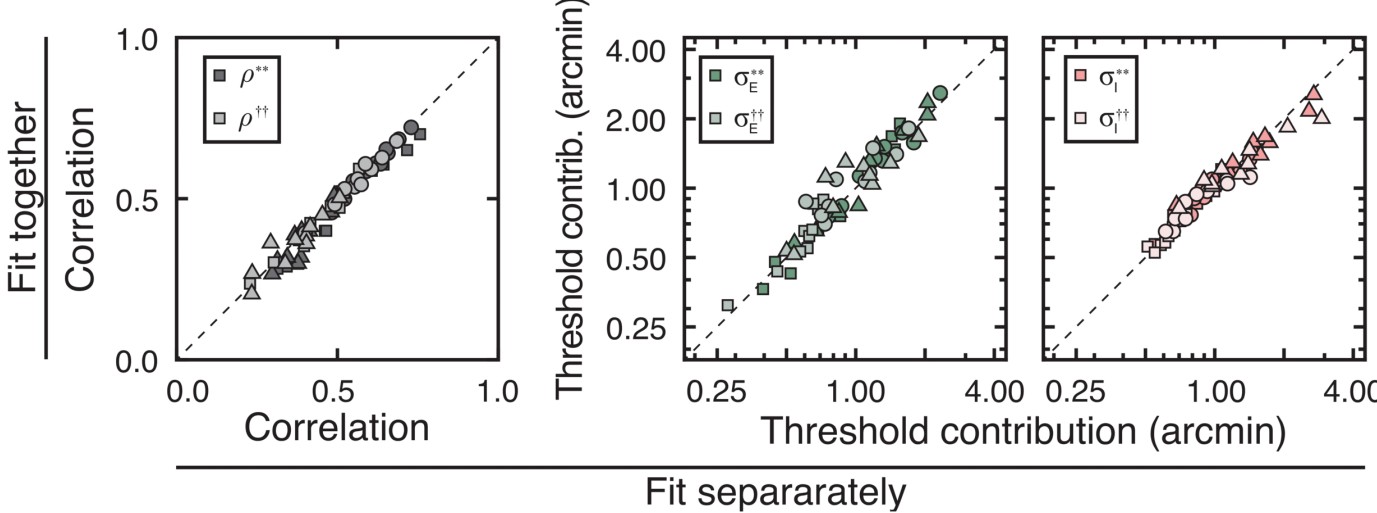

**Fig 10. Robustness of fitting methods.** Comparison of values obtained from fitting data from Experiments 1 and 2 separately (see Fig 6 and 9) versus together with a quasi-quadruple pass analysis. For decision-variable correlation (*left*), and threshold contributions by stimulus-driven factors (*middle*) and internally-driven factors (*right*), results are consistent regardless of the analytical approach. The consistency of the results indicates the validity and robustness of the quasi-quadruple pass analysis.

unaffected by the level of disparity-contrast (Fig 11A and 11B top row), the local-depth driven component has substantially higher variance with high than for with low disparity-contrast stimuli (Fig 11A and 11B bottom row).

These points are emphasized by histograms of the differences in the values of these components in the low and high disparity-contrast conditions. Although the luminance-pattern-driven component is essentially invariant to it (Fig 11B), the local-depth-driven component changes substantially with disparity-contrast (Fig 11D). From these results we conclude that the variance of luminance-driven component of the decision variable is a function of pedestal disparity but not disparity-contrast $\sigma_L^2(\delta_{\mathrm{std}})$, and that the local-depth-driven component is a function of both factors $\sigma_B^2(\delta_{\mathrm{std}}, C_\delta)$, a finding that strongly suggests that the components are not substantively affected by a potential common cause (e.g. local-depth variability). Overall, these results support the conclusion that natural luminance-pattern variability and natural local-depth variability in real-world scenes have separable effects on stereo-depth discrimination performance.

Note that the value of the interaction term is near-zero for all conditions (Fig 11C and 11D). Refitting the data with a model that fixes the interaction term to zero yields estimates of luminance-pattern- and local-depth-driven sources of variance ($\sigma_L^2$ and $\sigma_B^2$ respectively), and of the internal noise ($\sigma_I^2$), that are robust to whether the constraint on the interaction term is imposed during fitting (Fig 12; also see S2 Fig). Any qualitative description that applies to one set of fitted results applies to the other. Fits with and without the constraint also yield near-identical log-likelihoods. There is little evidence that non-zero covariances are required to account for the data.

Distinct features of natural scenes and images limit perceptual performance in distinct and largely independent ways. This is, perhaps, not surprising: local-depth variability is signaled by disparity-contrast, a stimulus feature that can be computed only by a binocular comparison of the eye's images, whereas luminance-pattern variability is monocularly computable. Their independence, however, is also not guaranteed. A common scene location

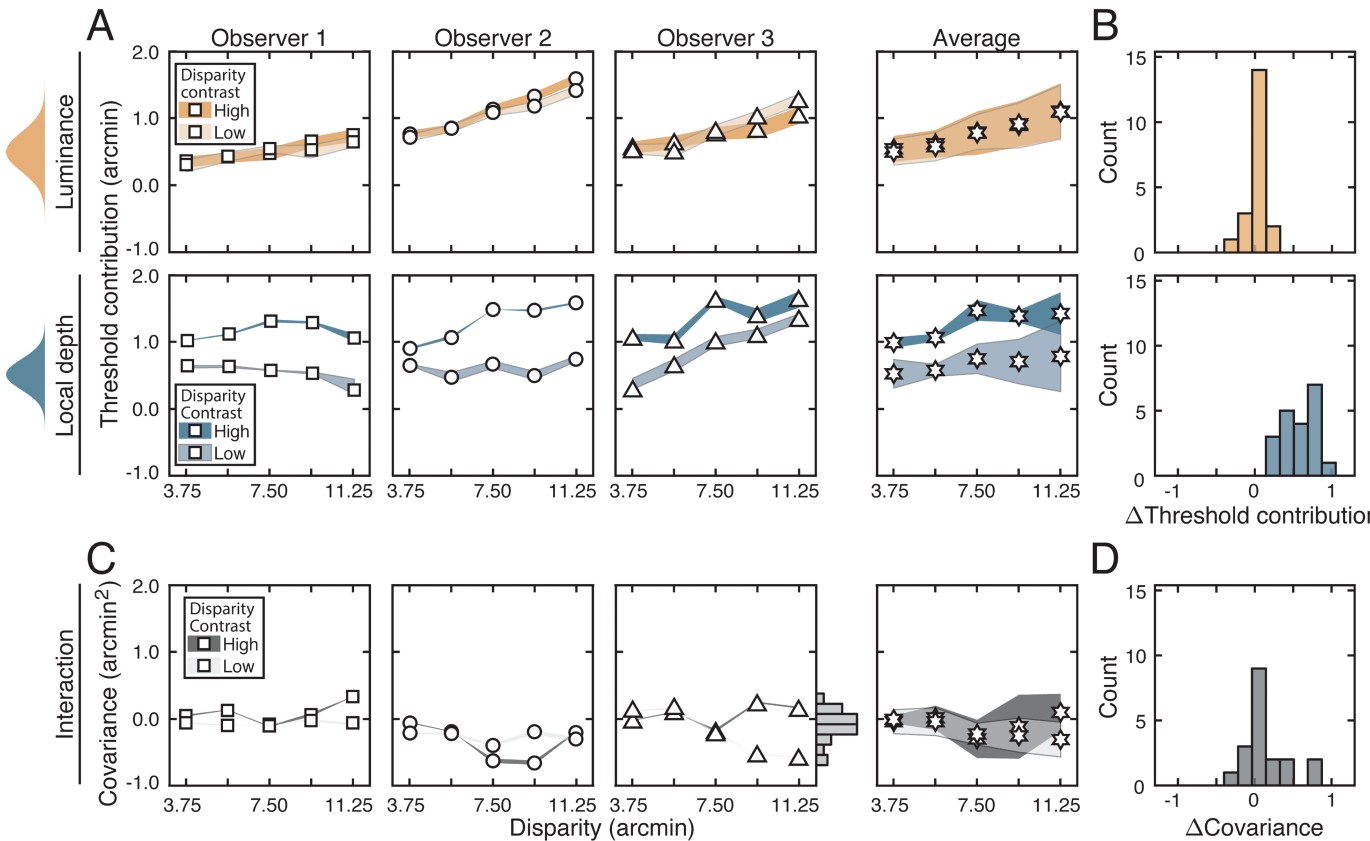

**Fig 11. Contributions of distinct stimulus-specific factors to thresholds, as revealed by the quasi-quadruple-pass analysis.** (A) Contribution of luminance-contrast pattern variability (*top*) and variability in local-depth structure (*bottom*) to threshold as a function of disparity pedestal at different disparity-contrast levels (*shades*), for each observer and the observer average. For individual observers, bounds of shaded regions indicate 68% confidence intervals for each condition, generated from 10,000 bootstrapped samples. For the observer average, bounds indicate across-observer standard-deviations. (B) Histogram of differences in luminance-pattern-driven and local-depth-driven threshold contributions across high and low disparity-contrast conditions, collapsed across disparity pedestals and individual observers. (C) Same as A, but for the interaction term (i.e. cov[*L,B*]). Histogram of the interaction terms collapsed across all disparity pedestals, disparity-contrasts, and individual observers is shown on the rightmost y-axis of the third column (mean=-0.11, sd=0.23). (D) Histogram of differences in the interaction term (i.e. cov[*L,B*]) across high and low disparity-contrast conditions, collapsed across disparity pedestals and individual observers. Data in C-D indicate that the interaction term is near-zero in all conditions.

gives rise to the luminance-contrast pattern (i.e. photographic content) in the left- and right-eye images, and to the pattern of binocular disparities between them. This might cause the effects of luminance-pattern variability to be correlated with those of local-depth variability; local regions with more depth variability could give rise to luminance (photographic/retinal) images with more variable luminance-contrast patterns, which could translate into correlated effects on performance. The data show that this is not the case. The results strongly suggest that each of these natural-stimulus-based sources of variability in the decision variable are near-independent of one another.

This result, when combined with other key results that have thus far been presented, provide a rich picture of the factors contributing to human disparity discrimination performance in natural scenes (Fig 13). Variability in luminance-pattern and variability in local-depth structure are independent external factors that limit performance. The effects of luminance-driven variation are the same regardless of the amount of local-depth variation in the scene. The local-depth-driven component of the decision variable is at its largest in the

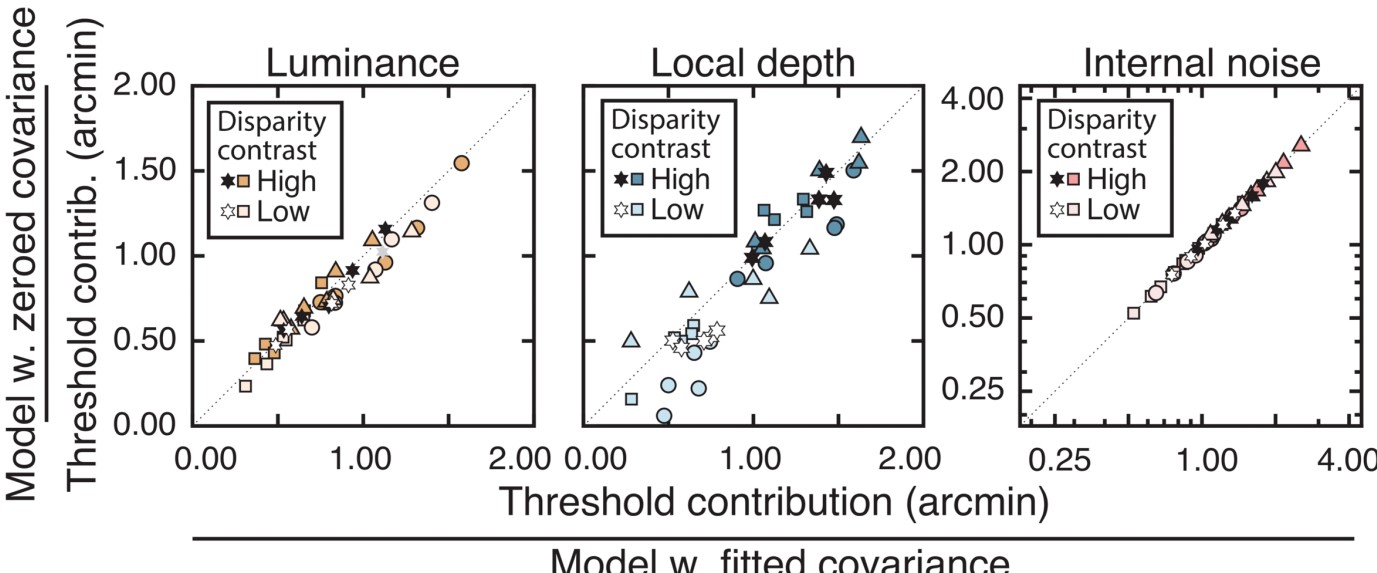

**Fig 12. Comparison between fitted values of luminance-pattern- and local-depth-driven variability in the quasi-quadruple-pass analysis when the covariances cov[L,B] are unconstrained versus when they are constrained to equal zero.** Fitted values for the luminance-pattern (*left*), local-depth (*middle*), and internal-noise components (*right*) remain largely consistent between the two fitting conditions. Different symbols represent different subjects. Stars represent across-subject averages.

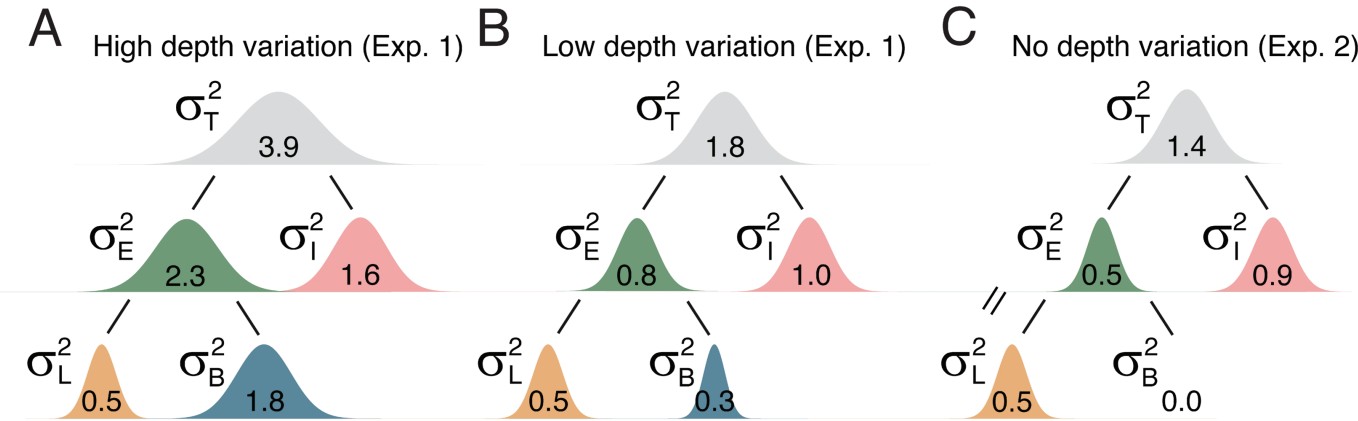

**Fig 13. Summary of main results.** The variance of the decision variable for disparity-based depth discrimination is highly dependent on the amount of local-depth variability in the scene. Total variability can be decomposed into externally stimulus-driven and internal-noise-driven components ($\sigma_E^2$ and $\sigma_I^2$) for each condition. The externally-driven component can be further decomposed into in factors of luminance variability ($\sigma_L^2$) and local-depth variability ($\sigma_B^2$). (Note that previous figures showed threshold contributions/standard deviations rather than variances; we show variances here for clarity, because they are additive.) When local-depth variability is *A*. high, *B*. low, and *C*. non-existent, $\sigma_B^2$ is the only external factor that changes appreciably. Internal-noise variance changes with external-noise variance. Numerical values indicate across-subject averages (see fourth column of Figs 5A, 6, 6A, 9, and 11) when the standard disparity was 7.500 arcmin. Averaging across all standard-disparities yields similar value.

high disparity-contrast condition (Fig 13A) and gets smaller as disparity-contrast is reduced (Fig 13B) or is eliminated entirely (Fig 13C). Hence, the local-depth-driven component of the decision variable is primary determinant of the performance differences in the different disparity-contrast conditions.

### Shared stimulus drive between observers

Earlier, we presented data showing that between-observer variability (i.e. threshold differences) was driven more by observer-specific differences in internal noise than by observer-specific differences in stimulus-driven variability (see Fig 7). We speculated that this result was due to a high degree of similarity between the computations that different humans use to extract useful information from each stimulus for the task. Here, we present data from between-observers decision-variable correlations that bolster the case.

Between-observers decision-variable correlation quantifies the similarity of the decision variable in two different observers across repeated presentations of the same stimuli. If different human observers are using the same computations to estimate and discriminate stereo-defined depth from natural stimuli, stimulus-by-stimulus disparity estimates from one human should be correlated with those from a second—that is, between-observers decision-variable correlation will be substantially larger than zero (assuming internal noise is not too large). On the other hand, if subjects are using quite different computations to process stimuli, stimulus-by-stimulus estimates or trial-by-trial responses from one observer will provide no information about estimates or responses from another, and between-subjects decision-variable correlation should equal zero.

We computed between-observers decision-variable correlation from response agreement data by straightforward adaptation of the quasi-quadruple-pass analysis (see Methods). However, because between-observers correlation is impacted by internal noise, its value does not transparently reflect the level of shared stimulus drive. The partial correlation does. Partial correlation is given by

$$\rho_{12 \cdot W} = \frac{\rho_{12}}{\sqrt{\rho_{11}\rho_{22}}} = \frac{\text{cov}[S_1, S_2]}{\sigma_{E1}\sigma_{E2}}, \tag{53}$$

where $\rho_{12}$ is between-observers decision-variable correlation, $\rho_{11}$ and $\rho_{22}$ are the within-observer decision-variable correlations, $S_1$ and $S_2$ are the stimulus-driven components of the decision variable that are shared between the two observers, and $\sigma_{E1}$ and $\sigma_{E2}$ are the standard-deviations of the stimulus-driven components of the decision variable in the two observers. This partial correlation provides more unvarnished information about what we are most interested in, because it is unaffected by internal noise. It quantifies the level of correlation in the stimulus-driven component of the decision variable between observers (see Methods).

Between-observers partial correlations are shown in Fig 14. Across all conditions and observer pairs, between-observers partial correlations are substantially above zero. In the high disparity-contrast conditions of Experiment 1, which are the conditions in which local-depth variability has its largest effects, between-observers partial correlations are 0.79 on average, with some values approaching the maximum possible value (i.e. 1.0). In the low disparity-contrast conditions of Experiment 1, the average value is 0.59. In Experiment 2, the average partial correlations for the high and low disparity-contrast conditions are 0.56 and 0.53, respectively (Fig 14 bottom row). Histograms of the differences between the high- and low-disparity-contrast conditions are shown in Fig 14B. And histograms of the raw values are shown in Fig 14C.

These results indicate that the majority—and, in one case (i.e. the high disparity-contrast stimuli with natural depth structure), the strong majority—of the stimulus-driven component of the decision variable is shared between observers. That is, natural stimulus variability

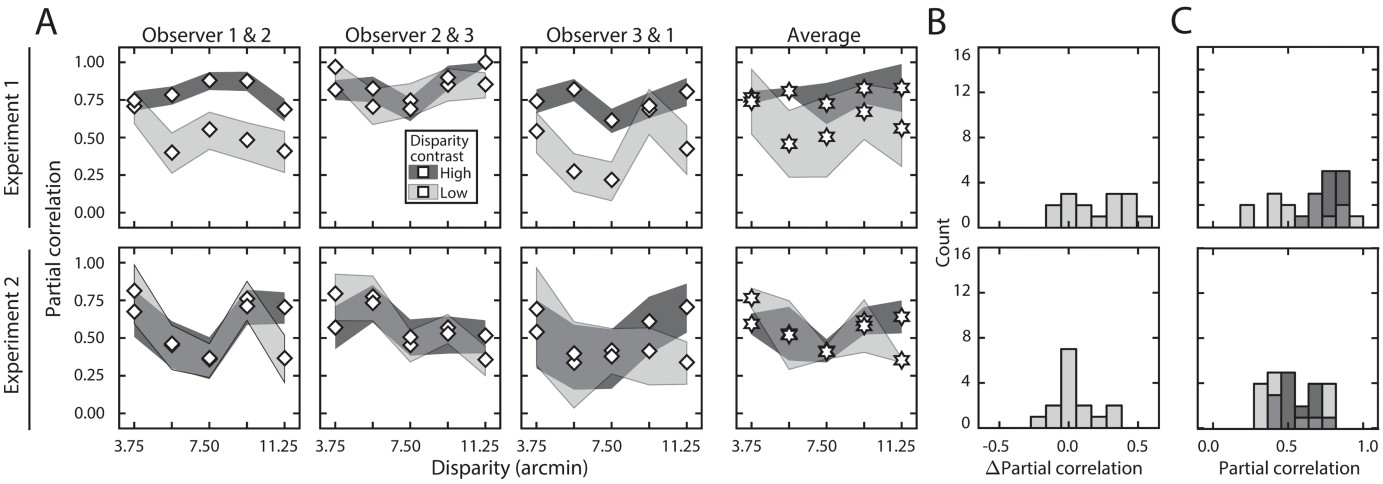

**Fig 14. Between-observer correlation in the stimulus-driven component of the decision variable, as revealed by the quasi-quadruple-pass analysis.** (A) Estimated partial correlation values, controlling for (i.e. removing) the effect of internal noise, between all observer pairs, for each experiment (*rows*), at all disparity and disparity-contrast levels. Averages across observer-pairs are shown in *column* 4. With the effect of internal noise removed, only the stimulus-driven component of the decision variable drives between-observer correlation. For observer pairs, bounds of shaded regions indicate 95% confidence intervals for each condition from 1,000 bootstrapped datasets. For the across-observer-pair average, bounds of shaded region indicates across-pair standard deviations. (B) Histogram of differences in partial correlation across high- and low-disparity-contrast conditions shown in A, collapsed across disparity pedestals and observer pairs. (C) Histograms of the raw partial correlations for each observer pair in A. In Experiment 1, the mean partial correlations are 0.79 and 0.59 in the high and low disparity-contrast conditions, respectively. In Experiment 2, the values are 0.56 and 0.53. The majority of the stimulus-driven variance is shared between observers.

associated with different stimuli having the same the latent variable (i.e. disparity) causes similar stimulus-by-stimulus over- and under-estimations of disparity-defined-depth in different humans. We conclude that the deterministic computations that the human visual system performs on individual stimuli are largely consistent across observers.

Chin and Burge [13], in the domain of speed discrimination, came to a similar conclusion using a related approach. By comparing human performance to that of an image-computable ideal observer, they found that differing levels of human inefficiency are near-exclusively attributable to different levels of internal noise. Like the current findings, this finding entailed that the variance of the stimulus-driven component of the decision variable is quite similar across different human observers, and is consistent with the visual systems of different human observers performing the same deterministic computations on the stimuli. The dovetailing evidence in stereo-depth and speed discrimination suggests that natural stimulus variability (natural variation in luminance patterns and/or depth-structure) has consistent effects on the visual systems of different human observers. These results suggest that evolution has honed the details of how visual systems compute so that they extract the most useful task-relevant information from natural stimuli.

## Discussion

In this article, using a natural-stimulus dataset, two double-pass experiments, and a series of analyses, we investigated human stereo-depth discrimination in natural scenes, with specific emphasis on how natural-stimulus variability limits performance. We sourced stimuli from a natural stereo-image database with a constrained sampling procedure, and computed ground-truth disparities directly from laser-range data at each pixel. Fixation (or pedestal) disparity, and local-depth variability—as quantified by disparity-contrast—were tightly controlled.

Luminance-contrast patterns and local-depth structures were allowed to vary naturally across the hundreds of unique stimuli that were sampled for each condition.

We find that the exponential law of disparity discrimination holds for human vision in natural scenes. We find that stimulus-driven variability and noise-driven variability have near-equal roles in setting these thresholds, and that the stimulus-based sources of variability make responses more repeatable (and thus potentially more predictable) across repeated stimulus presentations. We find that one of two underlying causes of the stimulus-driven variability is attributable to local-depth variation, multiplicatively increases discrimination thresholds, and is largely separable from luminance-contrast-pattern variation. And we find that different subjects make correlated stimulus-by-stimulus over- and under-estimations of disparity, suggesting that different human visual systems process individual natural stimuli with computations that are largely the same.

The approach developed here extends the rigor and interpretability that has been integral to progress in more traditional psychophysics and neuroscience experiments to more natural-stimulus sets [11,13,30]. In the real world, perceptual, and behavioral variability is driven by both external and internal factors. A comprehensive account of perceptual and behavioral variability, and the neural activity underlying it, must identify and describe the impact of all significant sources of performance-limiting variability. Encouragingly, the current results raise the prospect that an appropriate image-computable model may, in principle, be able to predict a substantial proportion of stimulus-by-stimulus variation across natural images.

## Progress and limitations

Progress in science is often incremental. Many times, it occurs by way of relaxing one experimental design element, while holding others fixed. We have investigated perceptual performance with stimuli sampled from natural scenes—which are atypical of laboratory experiments—while using conventional, tightly controlled, laboratory tasks [9,11,13,30]. Others have investigated performance with atypical tasks (e.g. free viewing and unconstrained eye-movements), while using conventional (e.g. Gabor) stimuli [1]. Both approaches have increased the ecological validity of the experimental conditions, and have provided new insights into the properties of neural computations underlying sensory-perceptual performance. But there are always limitations.

The stimuli used in the current experiments were foveally presented and subtended only 1° of visual angle, the approximate size of foveal receptive fields in early visual cortex. Foveal presentation of spatially-limited stimuli is common in psychophysical experiments, but doing so prevents the assessment of peripheral visual processing or how performance is affected by the dynamic interplay between eye, head, and body movements that occur in natural viewing. Limiting stimulus size to one degree also limits the extent to which contextual effects can affect performance. In the 'current experimentals, however, there was no evidence that one fixed spatial integration area accounted for performance any better than another (see S1 Fig). Experiments, possibly with larger stimuli, that are specifically designed to examine contextual effects could be interesting for future work.

Related issues concern the two-alternative forced choice (2AFC) procedure used in the current experiments. Although commonly employed, the rigid trial structure imposed by such designs is not well-aligned with how perceptual estimates, perception-driven decisions, and perception-guided action are inter-related in natural viewing. Alternative methods, such as continuous psychophysics, that more closely reflect the continuous interplay of perception and action in natural viewing, could complement the current findings [31–33].

Despite these limitations, the current experiments showed that the natural variation of luminance-contrast patterns and local-depth structures have large, distinct, and identifiable effects on performance. Developing methods that guide the judicious choice of stimulus sets and tasks that strike an appropriate balance between fully natural and tightly constrained, that are well-suited to available analytical methods, and are well-matched to the specific research question under study, will be increasingly important as the science becomes more focused on understanding how neurons respond and how perception works in the natural environment.

## Performance variation and prediction

An ultimate goal of perception science is to be able to predict, from an individual stimulus, the neural activity and subsequent perceptual estimate, whether it will be accurate or inaccurate, and whether it will be reliable or unreliable. The degree to which this goal is achievable hinges on the degree to which the stimulus-by-stimulus estimates are controlled by the properties of the stimulus, as opposed to noise. If the strong majority of performance variation is noise-driven, such efforts will be futile. So, before undertaking to develop and test models that make stimulus-by-stimulus predictions, it is prudent to demonstrate that a substantial proportion of performance variation is stimulus driven. In the current stereo-depth discrimination experiments, natural-stimulus-based sources of response variability account for approximately half of all performance-limiting variability (see Fig 5), a substantial proportion of which was shared across observers (see Fig 14).

However, while the stimuli—stereo-photographs of natural scenes—were allowed to vary naturally in many respects, the mean luminance was fixed to a comfortable photopic level, and luminance-contrast was set to the median contrast in natural scenes (see Methods) [4,34]. Both properties are known to impact stereo-depth discrimination performance [22], and stimulus detection performance in general [11,35–38]. Indeed, as mean-luminance and luminance-contrast increase, neurons respond more vigorously, signal-to-noise ratios increase, and performance becomes more reliable [34,39]. Hence, if luminance and contrast had been allowed to vary more naturally, the proportional contribution of stimulus-based factors to performance-limiting variability would likely have increased. The current estimates of stimulus-based contributions to the decision variable may therefore be underestimates of the total impact that stimulus-based factors would have in less tightly controlled circumstances. This speculation is supported by the fact that between-observers partial correlations are near the maximum possible values in the conditions in which natural stimulus variability was highest (see Fig 14).

The power of empirical datasets to help develop, constrain, and evaluate models can be improved by presenting unique stimuli on each trial. Many models can yield similar predictions of performance if only summary statistics (e.g. bias and precision) are used to evaluate the models' successes and failures. Image-computable models that predict decision-variable correlation and stimulus-by-stimulus estimates (or discriminations), in addition to bias and/or precision, can provide increased power for evaluating hypotheses about the neural activity and sensory-perceptual computations underlying performance [12,13,40].

## Stimulus-driven mechanisms

Our results show that stimulus variability is a major factor driving depth-discrimination performance. What are the mechanisms by which the visual system produces the specific patterns of trial-by-trial performance variation?

Binocular disparity is estimated by comparing image patches—and not isolated individual pixels—across the eyes, so some amount of spatial integration is undoubtedly occurring [41]. (The smallest known disparity sensitive receptive fields are approximately 6 arcmin in size [26,27].) Obligatory spatial integration necessarily reduces the reliability of solutions to the stereo-correspondence problem when local depth is varying (i.e. when disparity-contrast is high).

We tested whether spatial integration over a range of fixed-size areas could account for the trial-by-trial response variation. We found that no one fixed-size integration area provided more explanatory power than any another. Other simple mechanisms were similarly unable to account for the trial-by-trial response variation (see S1 Fig). It may be that mechanisms that dynamically adjust the integration area on a stimulus-by-stimulus basis are required to account for the trial-by-trial variation.

Ongoing computational research shows that probabilistic decoding of disparity from the responses of a model binocular receptive-field population is affected by stimulus variability in a manner similar to how it affects our human observers—that is, thresholds increase log-linearly with disparity and thresholds increase multiplicatively with disparity-contrast [42]. Interestingly, the probabilistic decoding routines employ a fixed strategy that implicitly adapts the spatial integration area on a trial-by-trial basis. Future work will be needed to determine whether similar mechanisms account for stimulus-by-stimulus variation in human performance.

## Noise and its impact on performance

In this article, we sought to partition the influence on performance of stimulus-driven from noise-driven variability, and to further partition the effects of two distinct types of natural-stimulus variability: luminance-pattern and local-depth variability. We made no attempts to determine different potential sources of noise (i.e. stimulus-independent sources of variability). As a consequence, any source of variance that led to less repeatable responses in the current experiments increased the estimate of noise variance. We conceptualized the noise as occurring at the level of the decision variable. But there are multiple stages in the chain of events preceding perceptual estimatation, both external and internal to the organism, where such variability could have originated and that would be consistent with the results.

Variation due to noise could have occurred during the initial encoding of the retinal image, in early visual cortex, at the decision stage (e.g. in the placement of the criterion), or a combination of these possibilities. Potential sources of such variation include the noisy nature of light itself [43], random fixational errors [44], neural noise [45,46], and trial-sequential dependencies [47]. Higher-level factors could also manifest as noise, including stimulus-independent fluctuations in alertness, attention, or motivation [48–51].

Experimental and computational methods that can determine the contribution of different types of stimulus-independent sources of variation are of interest to systems neuroscience [13,52]. There are clear steps that could be taken to identify and account for some of these potential sources of noise. Psychophysical methods have the potential to distinguish some of them. High-resolution eye-tracking would allow one to condition performance on the fixational state of the eyes [53–55]. Parametrically varying performance-contingent reward can systematically alter motivational state [51]. But neurophysiological methods would be required to identify and partition sources of noise internal to the nervous system that may arise at various stages of the visual processing and perceptual decision making pipeline. Paradigms that blend the advantages of the current approach for partitioning stimulus-based

variation with neurophysiological and computational methods for partitioning noise would be a useful way forward [30,52,56].

## External limits to human performance

Broadly construed, the current work continues in the tradition of the classic 1942 study of Hecht, Shlaer, and Pirenne. Its two most widely appreciated results are that, when fully dark adapted, (i) the absorption of a single photon reliably elicits a response from a rod photoreceptor and (ii) the absorption of five to seven photons in a short period of time reliably causes a reportable sensation of the light. Less widely appreciated is the finding that the limits of the human ability to detect light (i.e. light detectability thresholds) are attributable to the stochastic nature of light itself, a performance-limiting factor that is external to the organism. On a given trial at a given stimulus intensity, whether or not subjects reported that they had seen the stimulus depended near-exclusively on whether or not the requisite number of photons had been absorbed. That is, if the numbers of photons in proximal stimulus was identical, humans would respond identically. Performance was thus very tightly yoked to the variability of the external stimulus.

The results the current study suggest that, just as rod photoreceptors support performance in a very similar manner across different human observers, the computational mechanisms supporting the estimation and discrimination of depth in natural scenes are very similar across observers. In the current study, we showed that stimulus-based limits to performance become increasingly important as stimuli become ever more natural. If this pattern holds, it may be that stimulus-based limits to performance are by far the dominant factor as organisms engage with the natural environment. If true, image-computable models will have the potential to achieve strong predictive power from analysis of the stimulus alone. Such models, in which the underlying computations are made explicit, would have tremendous practical applications and deepen our understanding of how vision works in the real world.

## Supporting information

**S1 Fig**. **Variance in trial-by-trial response data explained by simple stimulus-driven mechanistic models**. Logistic regression was used to assess whether a number of different strategies could account for the variability in trial-by-trial responses in the high (*top row*) and/or low (*bottom row*) disparity-contrast conditions. (A) Variance accounted for by strategies that assume a fixed spatial-integration area as a function of integration diameter, where each disparity estimate is computed as the mean disparity within the integration area of each patch. Note that the largest integration diameter is equal to the area of the entire patch, and the smallest integration area was equal to the central region which had the same disparity value at each pixel up to a tight tolerance. In the former case, the disparity estimate equals the mean disparity of the patch. In the latter case, the disparity estimate equals the disparity of the central, target pixel. (B) Variance accounted for by strategies that assumed that the decision variable was determined by the largest near disparity (*nearest*), largest far disparity (*farthest*), and maximally deviant disparity (*max*) of each patch. (TIF)

**S2 Fig**. **Contributions of distinct stimulus-specific factors of luminance pattern and local-depth structure to assuming covariance cov$[L, B]$ is constrained to equal zero**. For reference, threshold-contributions fit without the zero-covariance constraint are also shown (*black-filled symbols*). Constraining the covariance has little effect for Observer 1, Observer 3, and the Average Observer. In Observer 2, the magnitudes of the threshold contributions are

systematically reduced, but the patterns are unaffected. In cases where black-filled symbols are not visible, they are plotted directly behind white-filled symbols. Contribution of luminance pattern variability (*top*) and variability in local-depth structure (*bottom*) to threshold as a function of disparity pedestal at different disparity-contrast levels (*shades*), for each observer and the observer average. For individual observers, shaded regions indicate 68% confidence intervals for each condition, generated from 10,000 bootstrapped samples. For the observer average (*last column*), shaded regions indicate across-observer standard-deviations. (TIF)

## Author contributions

**Conceptualization:** David White, Johannes Burge.

**Data curation:** David White, Johannes Burge.

**Formal analysis:** David White.

**Funding acquisition:** David White, Johannes Burge.

**Investigation:** David White.

**Methodology:** David White, Johannes Burge.

**Software:** David White.

**Validation:** David White.

**Visualization:** David White.

**Writing – original draft:** David White, Johannes Burge.

**Writing – review & editing:** David White, Johannes Burge.

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
