## [Decision Letter · Decision Letter 0]

18 Dec 2024

PCOMPBIOL-D-24-01213

How distinct sources of nuisance variability in natural images and scenes limit human stereopsis

PLOS Computational Biology

Dear Dr. White,

Thank you for submitting your manuscript to PLOS Computational Biology. After careful consideration, we feel that it has merit but does not fully meet PLOS Computational Biology's publication criteria as it currently stands. Therefore, we invite you to submit a revised version of the manuscript that addresses the points raised during the review process.

Please submit your revised manuscript within 60 days Feb 17 2025 11:59PM. If you will need more time than this to complete your revisions, please reply to this message or contact the journal office at ploscompbiol@plos.org. Please include the following items when submitting your revised manuscript:

We look forward to receiving your revised manuscript.

Kind regards,

Christoph Strauch

Academic Editor

PLOS Computational Biology

Marieke van Vugt

Section Editor

PLOS Computational Biology

**Additional Editor Comments :**

First of all, I want to apologize for the relatively long time it took until we found reviewers - as we can see from the reviews, this isn't due to the quality of the manuscript.

The reviewers and we agree that this is an elegant study that has the potential to advance the field. Overall, the requested methodological details, relatively simple analyses, and further discussion seem doable in a revision. The reviewers further suggested a couple of minor edits/visualizations that the authors may want to follow. Furthermore, scripts needed to reproduce the authors' findings should be made available.

**Journal Requirements:**

At this stage, the following Authors/Authors require contributions: David White, and Johannes Burge. Please ensure that the full contributions of each author are acknowledged in the "Add/Edit/Remove Authors" section of our submission form.

Potential Copyright Issues:

i) Please confirm (a) that you are the photographer of 1C, or (b) provide written permission from the photographer to publish the photo(s) under our CC BY 4.0 license.

**Reviewers' comments:**

Reviewer's Responses to Questions

Reviewer #1: Review of "How distinct sources of nuisance variability in natural images and scenes limit human stereopsis"

Reviewer: Michael Landy

This is a really interesting and well-done piece of work that comes to the conclusion that a surprising amount of variability underlying disparity judgments (of patches of natural images) is driven by stimulus variability rather than internal noise, and that the stimulus variability comes from two effectively independent sources (disparity pattern variability and luminance pattern variability). The data are convincing, the methods are perhaps partially drawn from previous work, but pushed to a new level in terms of the conclusions one can draw. All around impressive. But, the Methods section was quite a slog and could be made a bit easier to parse.

Comments, bigger and trivial (page/para/line):

6/3: I would hope the authors can share more than just data (e.g., analysis scripts, etc.).

7 et seq.: I'm still a little confused about the stimuli. The surround is meant to be at zero disparity with the image behind (and the "center" of the image at an uncrossed disparity that is one of the experimental parameters that is varied). The raised-cosine is kind of like a blurry or frosted window through which the scene is viewed. What is the disparity of the window's edge? I assumed it was at zero disparity, in which case more should be seen of the right eye's view at the left edge of the stimulus and vice versa. But, some of the text seems to indicate that a circular patch is cut from both eyes' views centered on the central corresponding point. Which is it, and please clarify that in the manuscript. If it's the latter, then I'm not sure how to think about the scene (mid-gray in front, but what happens at the edges to vignette the scene that's behind it?). I was worried that, e.g., if the standing disparity were 1 deg (and the former version of vignetting was used), there would be NO correspondence. Obviously, that's not what you meant. So, clarify the situation early, please. One confusion is in the definition of v_0, the vergence required to fixate the two center pixels. Maybe I missed it while reading, but I still wasn't sure when I got to this line that the center pixels corresponded.

Eq. 2: Does this pool both images? Using the same scaling?

8/2: Here is where you say the patches are centered on corresponding points, at last.

8/3/5: "central region": You should give its diameter here.

8/3/7: This would be better rephrased by giving the purpose first: "We are interested in the effect of disparity contrast on performance, so we chose patches whose disparity contrast fell either into a "low" range (details) or a "high" range (details), and we report performance below for each group of images separately."

8/4/3: I'm not sure it's cool to cite something as "n.d." with no source or means for a reader to find the citation.

10/2: I'd like a figure panel that shows the stimulus in situ with the background and "reticle" (whatever that is!). What's an "ordinal" direction? (New use of that word for me...)

11/5/2-3: 5x5x2 does not equal 10. Later, comparison disparity is NOT treated as a condition (but rather as an independent variable).

Eq. 4: d' is usual separation/spread, so why is it over variance here??? S/N is usually in energy terms, so separation^2/variance. This equation is neither.

13/1/5: Section 4.6 -> Eq. 6

Eq. 11: This tripped me up, because I stupidly looked at the equation and thought "r^2 is the proportion of variance accounted for, not r!" I was wrong, because this is not linear regression but the correlation between two variables with shared and exclusive variance, and that comes out differently. But, for readers who might be similarly tripped up by leaping to a false analogy, you might warn them. Maybe it's just me, though ;^/

Page 16, et seq.: This feel a bit like teaching rather than developing the math needed here. You are going to do a case that is symmetric and with a criterion at zero, but you develop the math for a general case with two different means (for the two passes) and nonzero criteria. Why bother? Once you assume zero and, in particular, equal criteria, and equal means for the two passes, the picture is symmetric (so that +- and -+ come out the same). It would be nice to save some verbiage by going with what is actually happening here from the get go. This will simplify Eq. 17, for example (using symmetry).

21/1/3: "depthnzz"

22/2/3: inconsistent WITH the expansion

22/2/11-12: twice "subscripts [to] denote" (delete "to")

24/2/1: with-observer -> within-observer

25/2/4: asses -> assess ;^)

25/4/3: Here's where you list 10 conditions the right way

Figure 2A: The Pass 2 timeline is missing the stimulus in the 2nd interval.

32/1/3: parallel to ONE another

32/2/1: nobody -> no one

33/3/6: You should probably show the data (as a function of pooling radius)

Eqs. 48 and 49: What's with all that space in the parentheses in the denominators?

41/3/4: These findings should also probably be shown in a figure.

Fig. 2B: The x-axis label is cut off

What does it mean that the dominant variability stimulus driven? It would be nice to speculate about this. You do ask whether they pool disparities around the center but for a larger area and reject that idea. But what if they use the largest far disparity? The largest near disparity? A randomly chosen small area on the stimulus? Some other strategy that gets knocked around by stimulus details and thus shows variability across stimuli? In other words, what does it mean that different stimuli produce different performance? Maybe the internal noise variance on a given trial depends, somehow, on the stimulus? You assume additive noise that is fixed, so what if it's Weberian based on something about the stimulus or otherwise multiplicative?

Reviewer #2: This manuscript introduces a method for partitioning the variability in a psychophysical task into stimulus-driven components (e.g., disparity and luminance contrast) and internal noise. The primary result, which is consistent with image-computable ideal observer analyses in past work by the same lab on different tasks, is that luminance variability drives perceptual variability substantially. Here, they find that disparity and luminance both contribute to perceptual performance and their contribution is correlated across observers. I think this type of variance decomposition is central to the scientific endeavor when we have “noisy” measurements or “noisy” behavior and the “Law of Total Variance” is a severely underappreciated conceptual framework in perceptual science. The challenge here is partitioning the variance with binary observations governed by truncated cumulative normal distributions. As far as I can tell, the authors got it right and the results are elegant. I have only a few comments.

My only major comment is that I wonder how different the results would be if you just put together a big logistic regression to predict every choice from the different measurable stimulus components and plot the Beta coefficients. It seems like the main advantage of portioning the variability as the authors did is that it naturally communicates the results in terms of effect size (i.e., what fraction of variance is due to what). I was surprised then that this quantity is reported in terms of “threshold contribution” with no units. My understanding is that these are capturing the variances? Why not represent in the “fraction of total variance”?

Optional clarity suggestions: it would be quite useful to see all the analytic solutions in one place (e.g., a table). I think the field would find this paper more useful if they could easily map the variance partitioning multi-pass experiment framework into their variables of interest. Right now, the presentation is thorough, but there’s a lot of jumping back and forth between equations to follow the logic.

Typos:

Equation 4: shouldn’t it be the standard deviation instead of the variance? For the formula in equation 3 to be correct, the denominator of d’ should be the common standard deviation. I believe this is probably a typo.

Section 4.9.3 : Depthnzz

Discussion “see Resutls”

Reviewer #3: This study applies an elegant logic to the task of separating out different components of variability that affect participants in a stereoacuity task using stereo photographs of natural stimuli. My main comments concern the clarity with which the authors describe the psychophysical task and stimuli. In an ideal world, they should be able to present a figure where the reader can see one of the main effects for themselves.

Major

1. Stimulus and task. It would help the reader to be clear what the stimulus was. p10 the Methods say 'In one interval... standard...In the other interval, ....comparison stimulus..' The assumption is that the same scene is presented in both intervals, with only the pedestal disparity changing. It would be helpful to make this explicit. The image in Fig 2A is too small to be helpful.

Also, for Fig 2A, delta is given as -7.50 arcmin. It might be helpful to change delta_{cmp} to something different to help the reader understand the stimulus.

Even more helpful would be for the reader to see two examples of a trial, so that they could see whether they get the main effect themselves. For example, the authors could repeat Fig 1C with a different pedestal disparity, then repeat both rows but now with a low disparity range. This should be considerably easier (Fig 4A). The icing on the cake would be a cross section of the stimulus in each case (x-axis showing pixel number across the image, y-axis showing disparity), times four (standard and comparison for high and low disparity contrast) so that readers instantly understands the stimulus and task. At the moment, a lot of reading of text is required to piece together the stimulus and task.

2. Monocular cues. If the same scene is used for the standard and comparison stimulus then, in theory, the task could be done monocularly (unless the authors sampled the scene with a slightly different lateral position of the scene relative to the camera to avoid this, but I did not find this in the text). This does not affect the authors' analysis (it would just contribute to lowering sigma_L, the luminance-driven variability), but it would be helpful if the authors could comment on this in the text.

3. Disparity range. I found it hard to pick what the authors' model was for the effect of disparity contrast (although they isolate the effect very clearly). For example, Harris and Parker (1995) discussed the efficiency of integrating disparity information across a number of samples in order to compare the mean disparity of dots spanning a range of disparities. Do the authors envisage a similar integration process? The authors point out that the participant could use the central pixel and perform in the same way as if the disparity contrast was zero, in which case no integration would be necessary.

4. A different experiment. It might be helpful for the reader if the authors mentioned in the Discussion what they thought would happen if the experiment had used an entirely different stereo photograph for the comparison stimulus. This would make the stimulus more similar to other experiments using dynamic random dot patterns where the luminance component is changed but the disparity profile remains. It would still be possible to do double pass experiments, provided the stimulus and comparison were repeated exactly on the second pass. As I understand it, the results of that experiment are not necessarily predictable from the sources of variability that the authors have gathered in the current experiment. Discussing this case would help to clarify how abstract a representation of disparity the authors believe participants retain between the standard and comparison (e.g., if just a single estimate of the pedestal value is retained, then changing between different stereo photographs might not matter).

Minor

p8. 'high' 'low' disparities, wrong way round.

p10. 'that proportion comparison...' Difficult sentence that does not effectively explain the basis on which the stimuli were chosen.

Fig 3. Explain the dashed line in the legend.

Fig 11. 'affect of' -> effect of

**Have the authors made all data and (if applicable) computational code underlying the findings in their manuscript fully available?**

Reviewer #1: **No: **

Reviewer #2: **No: **Data are available but code is not yet.

Reviewer #3: None

PLOS authors have the option to publish the peer review history of their article (what does this mean?). If published, this will include your full peer review and any attached files.

Reviewer #1: No

Reviewer #2: No

Reviewer #3: **Yes: **Andrew Glennerster

**Figure resubmission:**
---

## [Decision Letter · Decision Letter 1]

10 Mar 2025

Dear Dr White,

We are pleased to inform you that your manuscript 'How distinct sources of nuisance variability in natural images and scenes limit human stereopsis' has been provisionally accepted for publication in PLOS Computational Biology.

Best regards,

Christoph Strauch

Academic Editor

PLOS Computational Biology

Marieke van Vugt

Section Editor

PLOS Computational Biology

The reviewers have suggested only very minor improvements and noted a few small inconsistencies and language errors. All of these can be addressed during the proof stage, provided no further changes are made. Congratulations on this excellent paper!

Reviewer's Responses to Questions

**Comments to the Authors:**

Reviewer #1: This is an excellent revision. My comments are only typos/grammar SNAFUs (page/para/line):

6/5/5: chosen by random -> chosen randomly

8/5/7: consistent WITH the predictions

15/2/8: by the product OF the standard deviations

17/6/2-3: "Setting the decision variable..." is not a sentence!

18/1/1: "Hence" doesn't make sense. This is a conclusion from evidence that isn't provided in the previous sentences

Throughout: My standard pet peeve that bucks the tide: "data" is a plural word ;^)

19/1/14: Delete "*B-C*"

29/5/8: "Fig. Fig13B"

Some of the figures (e.g., Fig. 10) have been converted to lo-res pixels. Try to keep things as vector art later

Reviewer #3: Thank you to the authors for their very helpful replies and changes.

The new Fig 2 makes it much easier to understand the stimulus and task.

Fig S1 is also very helpful. As I understand it, the mean disparity of the stimulus is another potential cue (in addition to ‘largest near disparity’ or ‘largest far disparity’) that is not perfectly correlated with the nominal disparity of the stimulus? The disparity of the standard or comparison is defined by the disparity of the central region for that stimulus and that is not necessarily the same as the mean disparity of the stimulus. At least, that is how I understand the stimulus – if that is not the case, then more needs to be added to the section on ‘stimulus vetting’ to explain how this was done. Other studies have assumed that participants attempt to estimate the mean disparity when faced with a task like this. I do not think it would be fair to ask the authors for any more analysis at this stage, but a sentence somewhere, eg the legend for Fig S1B, would be helpful. It should mention that the mean disparity of the stimulus, like the largest disparity, is not perfectly correlated with the nominal disparity used in the figures.

It is now much clearer that the images used for the standard and comparison stimuli were different. In the legend for Fig 2, it says “Each natural image patch was unique across all trials and intervals.” This is not strictly true. It might help the reader to qualify this, e.g: “Over the course of a single pass, a unique natural image patch was used for every trial and interval.”

**Have the authors made all data and (if applicable) computational code underlying the findings in their manuscript fully available?**

Reviewer #1: Yes

Reviewer #3: Yes

PLOS authors have the option to publish the peer review history of their article (what does this mean?). If published, this will include your full peer review and any attached files.

Reviewer #1: **Yes: **Michael S Landy

Reviewer #3: **Yes: **Andrew Glennerster

---

## [Editor Report · Acceptance letter]

PCOMPBIOL-D-24-01213R1

How distinct sources of nuisance variability in natural images and scenes limit human stereopsis

Dear Dr White,

I am pleased to inform you that your manuscript has been formally accepted for publication in PLOS Computational Biology. Your manuscript is now with our production department and you will be notified of the publication date in due course.

With kind regards,

Anita Estes
